# The clinical and molecular landscape of breast cancer in women of African and South Asian ancestry

Graeme J. Thorn [1,3], Emanuela Gadaleta [1,3], Abu Z. M. Dayem Ullah [1], Lewis G. E. James[1], Maryam Abdollahyan [1], Rachel Barrow-McGee [2], Louise J. Jones[2] & Claude Chelala [1] ✉

Addressing existing racial disparity in breast cancer is crucial to ensure equitable benefit across diverse communities. We evaluate the molecular and clinical effects of genetic ancestry in African and South Asian women compared to European using a combined cohort of 7136 breast cancer patients. We find that non-European patients present significantly earlier and die at a younger age. The African group has an increased prevalence of higher grade and hormone receptor negative disease. The South Asian group shows tendency towards lower stage at diagnosis and tumour mutational burden. We observe differences and similarities in the somatic mutational landscape, and differences in germline mutation rates relevant to genetic testing and breast cancer predisposition. Potential therapeutic candidates are identified, with a higher propensity for homologous recombination deficiency serving as a therapy response indicator. We harness breast cancer multimodal data to improve understanding of ancestry-associated differences and highlight opportunities to advance health equity.

Breast cancer (BC) is the most commonly diagnosed cancer globally and the leading cause of cancer death in women, with an estimated 2.3 million new cases annually and accounting for about 685,000 deaths in 2020[1]. Racial disparities are observed in these figures, with higher mortality rates reported in ethnic minority populations. While social determinants of health have been shown to contribute to population-based differences in mortality, they do not fully explain the disparity observed[2].

Precision oncology is transforming the landscape of healthcare by offering a future in which the one-size-fits-all approach is superseded by tailored diagnosis and treatment. However, the under-representation of patients from ethnic minority populations in research studies and clinical trials limits the impact of translating these findings to non-white patients, exacerbating racial gaps in care delivery[3,4]. Observed disparities in outcomes between ethnic groups are likely perpetuated by differences in the distributions of germline pathogenic variants, the incidence of different subtypes, unique somatic mutations, pharmacokinetic behaviour, and tumour biology and behaviour[5–11]. Additionally, reports have shown that disease risk models and polygenic risk scores used for disease stratification can exhibit lower predictive accuracy in non-white populations[11–17].

Pan-cancer and disease-specific sequencing initiatives, such as– Genomics England 100,000 Genomes Project (100K GP)[18], The Cancer Genome Atlas (TCGA)[19], and Genes & Health (G&H)[20] are invaluable resources for research focussed on ethnic diversity. However, data derived from US-driven initiatives, such as the TCGA, consist of the dominant ethnic groups of their sampled cohorts; primarily White, Black and East Asian populations. As a result, these datasets may not capture the ethnic distribution or contributions of cohorts in studies conducted in other geographical regions. Furthermore, BC-specific

[1]Centre for Cancer Biomarkers and Biotherapeutics, Barts Cancer Institute, Queen Mary University of London, London EC1M 6BQ, UK. [2]Centre for Tumour Biology, Barts Cancer Institute, Queen Mary University of London, London EC1M 6BQ, UK. [3]These authors contributed equally: Graeme J. Thorn, Emanuela Gadaleta. ✉e-mail: c.chelala@qmul.ac.uk

initiatives, such as the Breast Cancer Now Biobank (BCN Biobank)[21], have proven an invaluable source of detailed clinical and molecular information that can be used to help advance research projects aimed at improving patient outcomes.

The Genomics England 100K GP[22], established in 2013, aimed to sequence 100,000 whole genomes from patients within the National Health Service (NHS) in England to facilitate incorporating genomic medicine into routine healthcare, delivered through 14 Genomic Medicine Centres across England. Collections were based on two themes: rare disease and cancer, with sequenced data being linked to its associated clinical data and then being made available to researchers within a dedicated trusted research environment (TRE). The TCGA[23] is a comprehensive programme comprising molecular and clinical data from over 20,000 samples spanning 33 cancer types. The multi-omics data generated from sequencing and array-based technologies is publicly available to the research community for use in projects as a research or validation dataset[24]. G&H[25] is a community-based general health study that recruits individuals of Pakistani and Bangladeshi origin within the East London, Bradford and Greater Manchester areas, with germline genomic and clinical data available within its own TRE. These participants are consented for lifelong access to their primary and secondary electronic healthcare records (EHRs) and a saliva sample for genetic studies. Finally, the BCN Biobank[24] is the UK's largest unique BC disease-specific collection of high-quality specimens, comprising tissues, serial liquid biopsies and bespoke cell lines, and longitudinal clinical data derived from primary and secondary EHRs.

In this work, we harness multimodal data available from four cohorts–Genomics England 100,000 Genomes Project, The Cancer Genome Atlas, the Breast Cancer Now Biobank and Genes & Health–to improve our understanding of ancestry-associated differences in BC and highlight opportunities to address inequalities and achieve more equitable clinical outcomes (Fig. 1). The collection of data in our study represents one of the largest UK South Asian BC cohorts currently available.

## Results
### Cohort characteristics
The study dataset comprised 7253 BC patients from four cohorts– 3334 from Genomics England, 2479 from Barts Health NHS Trust patients within BCN Biobank (BCN Biobank-Barts), 1076 from TCGA and 364 from G&H (Fig. 2). To minimise biological variability between the cohorts, the inclusion criteria were restricted to female BC patients presenting with primary disease. For Genomics England and TCGA, the analytical cohort was further restricted to those patients for whom their primary tumours were sequenced and genetic ancestry (gAncestry), determined from germline sequencing, was available. For G&H, only the gAncestry was available from germline sequencing. For the BCN Biobank-Barts cohort, sequencing data, and gAncestry, were available for 231 patients that were dually consented to BCN Biobank and Genomics England. For the rest of this cohort, self-reported ethnicity (SRE) was used.

For Genomics England, 2840 patients passed initial inclusion criteria, of whom 2781 also had gAncestry available (see Methods). This analytical cohort comprised European (EUR, $n = 2343$ (84.3%)), African (AFR, $n = 138$ (5.0%)), South Asian (SAS, $n = 123$ (4.4%)), East Asian (EAS, $n = 37$ (1.3%)), American (AMR, $n = 12$ (0.4%)) and Admix (Admix, $n = 128$ (4.6%)) populations (Supplementary Fig. 1a). The gAncestry profile of the TCGA analytics cohort ($n = 1064$) differs in its composition of Asian gAncestry groups to Genomics England and comprises EUR ($n = 821$ (77.2%)), AFR ($n = 125$ (11.7%)), SAS ($n = 4$ (0.4%)), EAS ($n = 56$ (5.3%)), AMR ($n = 5$ (0.5%)) and Admix ($n = 125$ (11.7%)) gAncestry populations (Supplementary Fig. 1b). As G&H is an initiative that looks at more general health concerns within communities of South Asian British heritage, the participants without a history of BC and without

inferred gAncestry information were excluded from initial analyses (56,707 out of 57,047), leaving 340 patients with an inferred gAncestry of SAS in the analysis cohort (Supplementary Fig. 1c).

Following initial filtering for women with primary disease, the ethnic distribution of the BCN Biobank analytical cohort ($n = 2126$) was as follows: White ($n = 1332$ (62.7%)), Black/Black British ($n = 321$ (15.1%)), Asian/Asian British ($n = 234$ (11.0%)), Other Ethnic group ($n = 142$ (6.7%)), Mixed ($n = 56$ (2.6%)), and not stated/unknown ($n = 41$ (1.9%)) (Supplementary Fig. 2).

Patients within the AMR and EAS gAncestry groups were removed from the final analytical cohorts due to the small sizes of the cohorts, which would significantly reduce the power of any statistical inference on this cohort. Furthermore, the limited number of participants in the TCGA SAS population ($n = 4$), precludes the use of this population as a validation for SAS v EUR comparisons. The clinical and molecular analyses in this study focus on the EUR, AFR and SAS gAncestry superpopulations within all cohorts.

To determine the ethnic representativeness of the cohorts relative to the geographical region of collection, SRE was compared to that of the UK Census data for Genomics England and the BCN Biobank-Barts cohort. The ethnic composition of the Genomics England cohort broadly recapitulates that of England (2021 UK Census data–Supplementary Fig. 2) with 81.0% White, 4.2% Black/Black British, 9.6% Asian/Asian British, 3.0% Mixed, 2.2% Other. However, it is not comparable to that of London, which comprises a higher proportion of ethnic minority groups (53.8% White, 13.5% Black/Black British and 20.7% Asian/Asian British). The BCN Biobank-Barts cohort closely resembles the London statistics, likely representing the ethnic composition of the area around the key collection site in North East London (Supplementary Fig. 2a). The gAncestry composition of the TCGA analysis cohort (Supplementary Fig. 2b) presents a higher proportion of AFR gAncestry and a much lower proportion of SAS gAncestry relative to Genomics England (AFR, 11.5% in TCGA v 5.0% in Genomics England; SAS, 0.4% TCGA v 4.4% Genomics England) recapitulating the diversity of the geographically diverse TCGA data collection centres.

### Concordance between self-reported population descriptors and gAncestry
There is high, almost identical, concordance between population descriptors and gAncestry in Genomics England (92.9%) and TCGA (92.3%) (Supplementary Fig. 1). Concordance between the BCN Biobank SRE and gAncestry for dually-consented patients is 88.5%. However, this figure improves to 94.5% if the Discovery Data Service[26] is used to improve stated ethnicity. G&H exhibited the highest concordance between SRE and gAncestry (99.7%), however this was expected due to the recruitment criteria of this cohort (Supplementary Fig. 1c).

Patients assigned to the Admix cohort in Genomics England represent the greatest source of discordance between SRE and gAncestry, with 50% concordance between a mixed SRE and Admix gAncestry (Supplementary Fig. 1).

### Descriptive statistics of clinical and molecular features
We explored the associations between gAncestry and key clinical or molecular variables by applying linear regression for numeric variables (such as age at diagnosis and age at death), logistic regression for categorical variables (such as receptor status, tumour stage and grade) and a scaled negative binomial regression for the tumour mutational burden (TMB). The group comprising the largest number of individuals, EUR, was used as a reference against which all other gAncestry groups were compared (Supplementary Methods).

The results for Genomics England indicate that AFR and SAS women present to the clinic with BC significantly earlier (5.28 years [95% CI (3, 7.55), linear regression $p = 5.7 \times 10^{-6}$] and 6.91 years [95% CI (4.52, 9.3), linear regression $p = 1.7 \times 10^{-8}$] respectively) and die at a

## a

### UNIQUE CLINICAL & SEQUENCING COHORTS

## b

### DATA ANALYTICS AND LINKAGE

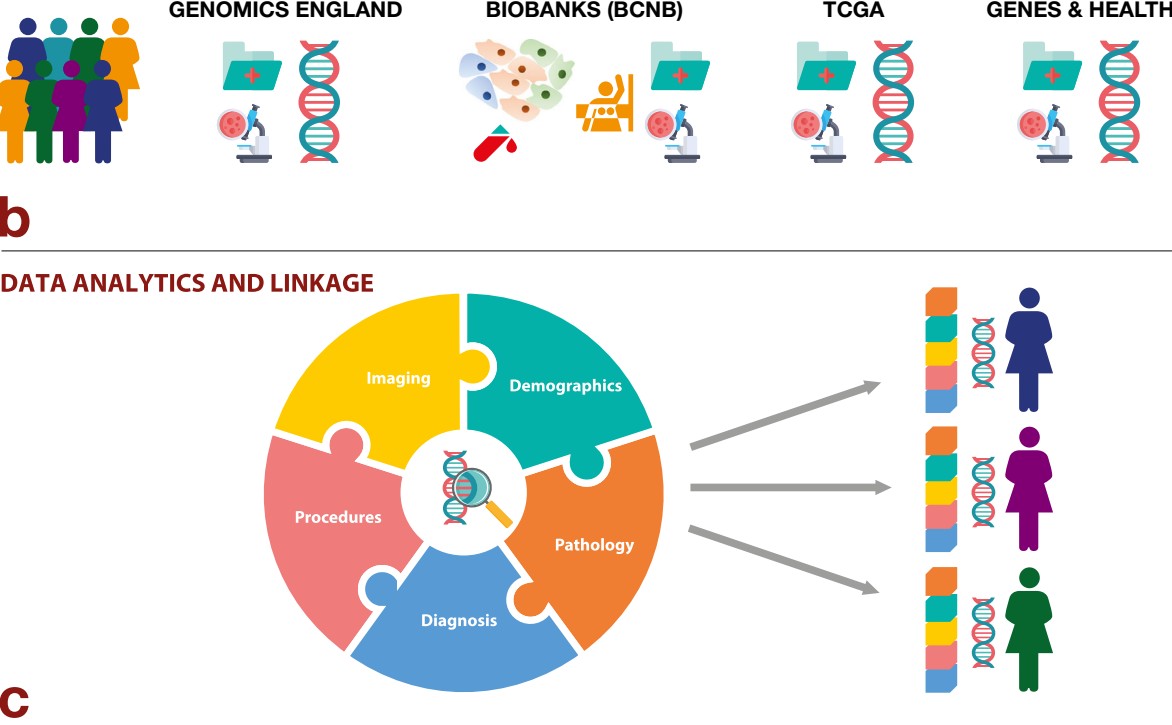

## c

### MOLECULAR AND CLINICAL PHENOTYPING

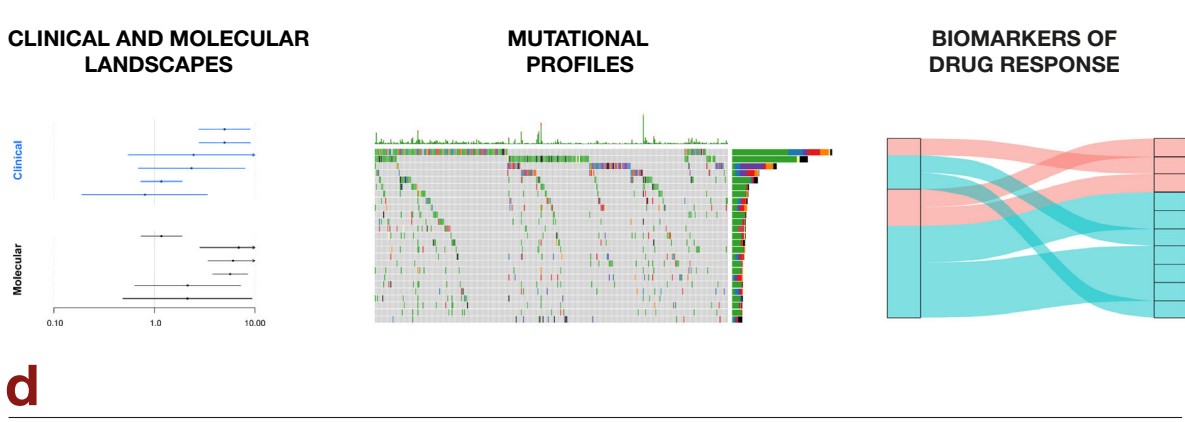

## d

### GUIDE STRATIFIED CARE BASED ON CLINICAL JOURNEY AND INDIVIDUAL GENETIC MAKEUP

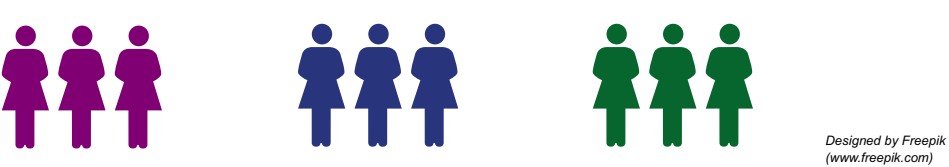

*Designed by Freepik
(www.freepik.com)*

**Fig. 1 | Graphical summary of the analytical approach. a** The four multi-modal clinical and sequenced cohorts used in this study. **b** Data analytics and linkage, focusing on developing a framework for harmonising and linking a defined set of EHR data and deriving a research-ready dataset alongside the sequencing data. Stratification is performed based on the threshold-inferred genetic ancestry of patients. **c** Data processing workflow to explore clinical data alongside sequence data. For a given genetic ancestry, we explore the relationship between BC clinical variables, mutational profiles and regulators of drug response. **d** This framework supports the use of multi-modal longitudinal EHRs and sequencing in BC to inform care for under-represented superpopulations. This figure uses icons designed by Freepik (www.freepik.com).

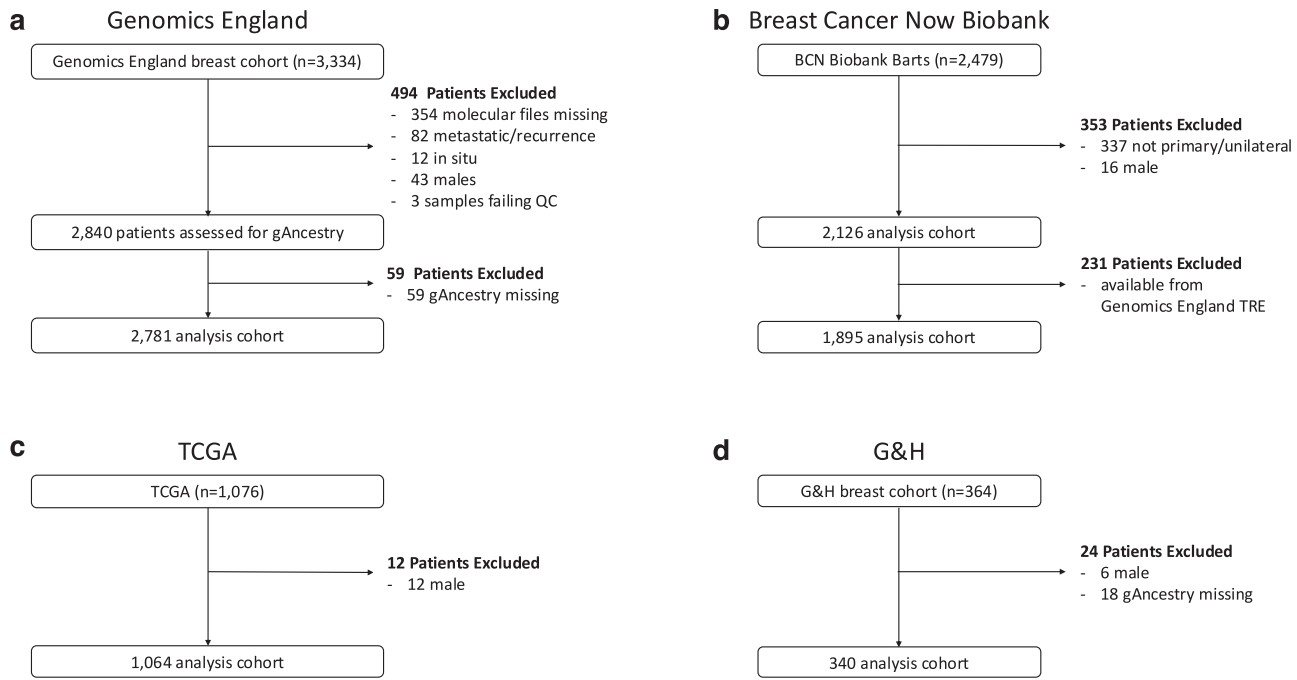

**Fig. 2 | Schema of the four cohorts in this study. a** Genomics England, **b** BCN Biobank-Barts cohort, **c** TCGA, **d** G&H. For each cohort the inclusion and exclusion criteria are labelled, leading to final cohort sizes of Genomics England ($n = 2781$), BCN Biobank-Barts cohort ($n = 1895$), TCGA ($n = 1064$) and G&H ($n = 340$).

younger age (8.94 years [95% CI (2.91, 15), linear regression $p = 0.0039$] and 13.20 years [95% CI (5.79, 20.7), linear regression $p = 5.5 \times 10^{-4}$]) earlier, respectively) than their EUR counterparts (Fig. 3a, Supplementary Data 1). Patients in the AFR population also present with higher grade tumours (OR 1.88 [95% CI (1.3, 2.71), logistic regression $p = 7.2 \times 10^{-4}$]), with a higher incidence of hormone receptor negative (HR-) disease relative to EUR (ER- OR 2.06 [95% CI (1.29, 3.28), logistic regression $p = 0.0024$]; PR- OR 2.07 [95% CI (1.3, 3.29), logistic regression $p = 0.002$]). Similar findings were reported in the AFR group of the TCGA, with these patients presenting at a younger age (4.08 years earlier, [95% CI (1.58, 6.58), linear regression $p = 0.0014$]) and with increased HR- disease compared to the EUR group (ER- OR 2.9 [95% CI (1.92, 4.37), logistic regression $p = 3.9 \times 10^{-7}$]), PR- OR 2.37 [95% CI (1.6, 3.51), logistic regression $p = 1.7 \times 10^{-5}$]) (Supplementary Fig. 4a, Supplementary Data 2). There is a tendency for AFR patients to present with slightly lower ancestry-corrected TMB relative to their EUR counterparts, with this being significantly lower in the SAS population (effect size 0.845 [95% CI (0.733, 0.975), scaled negative binomial regression $p = 0.02$]).

A relationship was observed between gAncestry and quintiles of index of multiple deprivation (IMD) within Genomics England, with AFR and SAS patients more likely to reside in areas within the more deprived quintiles, an association that is more pronounced in the former (Fig. 3a, Supplementary Data 1). Tests to determine whether IMD was confounding the analyses for the other variables showed no significant difference when adding IMD as the second variable (Supplementary Data 3), bar the age at diagnosis (likelihood-ratio test $X^2 = 14.0$ (4 d.f.), $p = 0.007$). A test of clinical variables versus IMD alone for the largest gAncestry group (EUR) showed that these patients in the most deprived quintile present 2.38–3.12 years earlier than those in the other four quintiles, where age at diagnosis was similar. No other factor was significantly associated with IMD (Supplementary Data 3, Supplementary Data 4).

Associations between these clinical variables and SRE were investigated in the BCN Biobank dataset. In agreement with our Genomics England gAncestry-based findings, Black and South Asian patients in the BCN Biobank presented earlier (2.54 years [95% CI (0.932, 4.15), linear regression $p = 0.002$] and 2.75 years [95% CI (0.898, 4.59), linear regression $p = 0.0036$], respectively) and died at a younger age (6.48 years [95% CI (2.39, 10.6), linear regression $p = 0.002$]), and 6.21 years [95% CI (1.16, 11.3), linear regression $p = 0.016$], respectively) relative to their White counterparts (Supplementary Fig. 4b, Supplementary Data 5), although the difference between the populations is smaller. Examining the IMD distribution from dually consented EUR patients within Genomics England shows that this particular subset derives from areas of higher deprivation, which may account for the reduction in effect size if this occurred across the whole BCN Biobank. Furthermore, patients within these ethnic groups also tended to present with aggressive–significantly higher frequencies of high-grade tumours and lymph node involvement–HR- disease.

We examined each gAncestry group within Genomics England split based on a 50-years-old cut-off, to represent the age at which the NHS breast screening is initially offered to individuals, however this resulted in small numbers of participants in these sub-groups, increasing the uncertainty in our point estimates (Supplementary Fig. 5a).

Patients within the <50-year-old AFR group show a propensity to present with higher grade tumours, HR- disease and a HER2+ receptor status. In addition, the TMB of this younger cohort tends to be higher than their EUR counterparts (Supplementary Fig. 5a). The clinical features of the <50-year-old SAS group are like those of the corresponding EUR group, bar a potential trend towards PR- disease, a trend that appears inverted in the ≥50-year-old group. Further examination of TMB trends stratified by ER status across the cohort identified higher median TMB in ER- patients (2.54 muts/Mb) compared to ER+ patients (1.33 muts/Mb; $p < 2.2 \times 10^{-16}$, two-sided Wilcoxon rank-sum test). Finally, as observed in the unstratified analysis, non-EUR patients in the ≥50-year-old cohort died at a younger age (5.80 years [95% CI (0.28, 11.3), linear regression $p = 0.04$] and 9.47 years [95% CI (2.69, 16.3), linear regression $p = 0.0066$] earlier for AFR and SAS, respectively) (Supplementary Fig. 5a).

The age at diagnosis curves for each gAncestry show differences between the three populations. The EUR cohort distribution curve of age at diagnosis in the Genomics England analytical cohort is flatter relative to the non-EUR groups, with

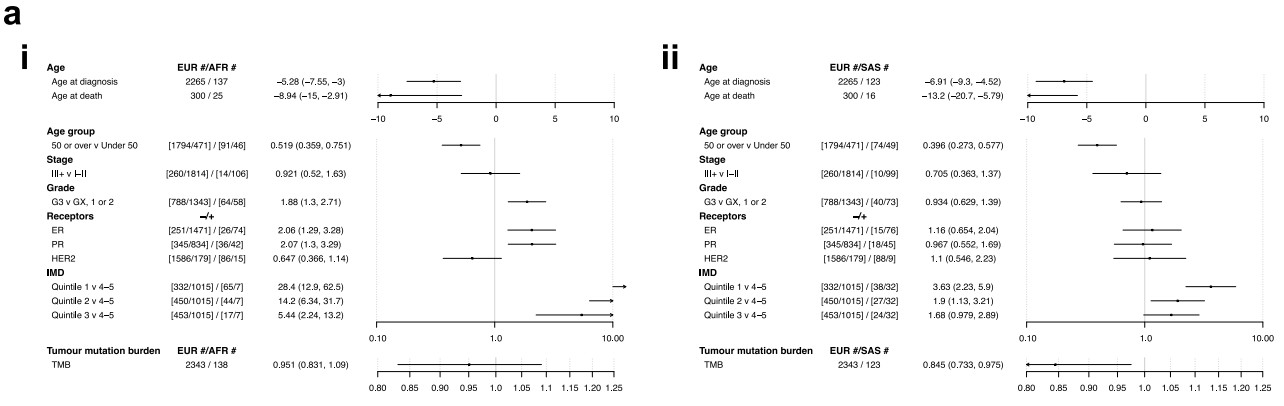

**Fig. 3 | Clinical and demographic details of the four cohorts. a** Forest plots of the clinical and molecular features of the non-EUR cohorts relative to EUR in the Genomics England cohort: ORs and 95% confidence intervals shown for linear (ages), scaled negative binomial (tumour mutation burden) and logistic regressions (other clinical details). **b** Age of diagnosis distributions for EUR (Genomics England/ TCGA)/White (BCN Biobank), AFR (Genomics England/TCGA)/Black (BCN Biobank), and SAS (Genomics England))/Asian (BCN Biobank and G&H). **c** Visualisation of the gAncestry-derived screening windows in the Genomics England cohort.

two modes of similar heights at 51 and 70 years (Fig. 3b, Supplementary Fig. 5b). There is a unimodal age distribution for patients of AFR and SAS gAncestries, with prominent peaks at 47 and 49 years, respectively.

The age distribution of the EUR cohort of the TCGA exhibits a mild bimodal distribution, with the highest peak presenting at 63 years and the mean age at diagnosis being similar to that of the Genomics England analytical cohort (59.42 years and 61.7 years, respectively) (Fig. 3b,

Supplementary Data 2). The White cohort from BCN Biobank, whilst showing a larger IQR than non-White cohorts, appears to shift towards a younger age at presentation (58.41 years) (Fig. 3b, Supplementary Data 5). As mentioned, this could be attributable to the BCN Biobank recruitment centre being located close to areas of higher deprivation in North East London.

Similar trends in age distribution profiles are observed between the AFR/Black cohorts of Genomics England, TCGA and BCN Biobank, and the SAS cohorts of Genomics England and G&H (Fig. 3b, Supplementary Data 2 and 5). The South Asian cohort of the BCN Biobank exhibits a wider peak than that from both Genomics England and G&H, although the mean age of diagnosis is similar (BCN Biobank 55.2 years; Genomics England 54.8 years). This may be due to the differences between ethnic subgroups within BCN Biobank: Bangladeshi patients ($n = 82$) had a mean age at diagnosis of 53.99 (s.d.= 11.94 years); Pakistani patients ($n = 65$) had a mean age of 54.37 (s.d. = 12.54); but the Indian patient group ($n = 87$), had a higher mean age of 57.20 (s.d.= 10.03).

We stratified each gAncestry cohort within Genomics England based on the central 60% of the age distribution, to improve equity between the cohorts in breast screening. Current guidelines apply at the 20th percentile in the EUR age distribution, so we placed the lower bound on the intervals at that percentile in the other gAncestries. We also increased the upper bound to the upper 20th percentile of the age distribution. This resulted in gAncestry-specific intervals: 50–75 years (EUR), 47–69 years (AFR) and 45–68 years (SAS) (Fig. 3c). With gAncestry not readily available in the clinic, we isolated patients with concordant SRE and gAncestry assignments and confirmed the applicability of these modified windows (Supplementary Fig. 5b). Applying the screening windows to BCN Biobank patients not dually-consented with Genomics England, gives better coverage for all three major ethnic groups: 67.4% (793/1176) within White patients, 64.8% (190/293) within Black patients and 70.5% (148/210) within Asian patients, and applying the suggested Asian age window to the G&H cohort would improve coverage from 26.4% (96/363) for the standard 50-70 window to 59.2% (215/363) within this group.

## gAncestry-associated somatic variants within the Genomics England cohort

The variant landscape of the TCGA cohorts has been described previously[6]. To determine gAncestry-associated somatic variant profiles within the Genomics England cohort, two models were applied to identify differentially mutated genes and variants relative to the EUR comparator: logistic regression (all gAncestries jointly, with EUR as reference), and a threshold criterion. The latter was implemented to identify variants exclusively present in one cohort, where the application of a logistic regression model would fail to converge (Supplementary Methods).

Significant differences in the mutational frequencies of 481 genes in AFR group and 275 genes in the SAS group were observed (Supplementary Data 6). Of these, the logistic classifier identified 2 and 19 genes (Fig. 4) and the threshold classifier 479 and 269 genes in the AFR and SAS cohorts respectively (Supplementary Data 6). Six genes (*RBMS*, *OTOF*, *FBXW7*, *NCKAP5*, *NOTCH3* and *GPR158*) were found commonly differentially mutated between the AFR and EUR populations of the Genomics England and TCGA cohorts, with *RBMS* also enriched in the SAS gene list for Genomics England.

We identified 71 variants with significant differences in mutational frequencies in the AFR cohort and 60 variants in the SAS cohort (Supplementary Data 7). Of these, the logistic classifier identified 7 and 4 variants (Fig. 4) and the threshold classifier 70 and 52 variants in the AFR and SAS cohorts respectively (Supplementary Data 7).

Running the logistic regression on AFR and SAS populations (excluding the EUR population) determined that the computed effect sizes were equal to the differences between the EUR and non-EUR comparisons, resulting in no novel observations.

## Germline variant profiles of cancer susceptibility genes

We applied logistic regression, using the EUR population as baseline, to identify gAncestry-associated germline variants within the Genomics England (EUR, AFR and SAS) and the TCGA (EUR and AFR) cohorts. This analysis focussed on variants within 180 germline genes: those tested in the clinic (as reported in the NHS Genomic Medicine Service Panels v80)[27]; and those conferring susceptibility to cancer (as reported in the using the PanelApp gene panel[28]). Variants within these genes were prioritised based on the level of evidence supporting a ClinVar classification of pathogenicity (see Methods). The same analytical parameters were applied to germline variants in the G&H cohort to allow for case:control observations within the SAS population.

Differences in the prevalence of germline variants of cancer susceptibility genes were observed between the gAncestries (Fig. 5a, b, Supplementary Data 8). The mean number of both pathogenic variants and those of unknown significance (VUS) per participant was higher in the non-EUR populations, with participants in the AFR group exhibiting the greatest number of variants in both classifications (6.45 pathogenic variants per participant and 21.23 VUS per participant) relative to the reference EUR population (5.27 and 19.54, respectively).

Of the pathogenic ($n = 37$) and VUS ($n = 43$) variants identified as significantly mutated between EUR and AFR populations in Genomics England, a 36.3% overlap was observed with those reported in the TCGA between the same populations (pathogenic, 45.9%; VUS 27.9%) (Fig. 5c, Supplementary Data 8). Only one of the 23 pathogenic variants identified as significantly mutated between SAS and EUR populations, *BRCA1*, was also identified in the case:control study using G&H, supporting that this gene confers susceptibility to BC (Supplementary Data 8).

The AFR and SAS groups in Genomics England were both significantly enriched for *BRCA1* (AFR, OR 6.21 [95% CI (2.72, 14.1), logistic regression $p = 1.4 \times 10^{-5}$]; SAS, OR 4.27 [95% CI (1.6, 11.4), logistic regression $p = 0.0038$]) mutations as well as those commonly associated with mechanisms of DNA damage: nucleotide excision repair; mismatch repair; Fanconi anaemia germline mutations (Supplementary Data 8). In addition, the AFR population also exhibited a higher frequency of additional variants associated with breast cancer predisposition: *BRCA2* (OR 2.24 [95% CI (1.31, 3.84), logistic regression $p = 0.0031$]) and *PALB2* (OR 2.92 [95% CI (2.04, 4.19), logistic regression $p = 5.8 \times 10^{-9}$]).

To test whether ER status could be confounding this analysis specifically for germline *BRCA* mutations, we split the Genomics England analytic cohort by ER status and found a small difference between ER- disease (7.5% of patients, $n = 22/292$) and ER+ disease (7.1% of patients, $n = 116/1621$), which was not significant (Fisher's exact test $p = 0.8061$).

## Mutational signatures and homologous recombination deficiency

To determine the contribution of environmental exposures to differences in mutation frequencies between the gAncestry groups within Genomics England, we implemented the signature.tools.lib R library[29] to perform mutational signature analyses confined to breast-specific parameters.

Our dataset comprised 32,558,096 substitutions, 339,207 double substitutions, 15,384,289 indels, and 489,339 rearrangements (Table 1). There are significantly more substitutions, double substitutions, and rearrangements per patient in the AFR group, and fewer indels, whereas the SAS group tended to mirror this trend with fewer substitutions, double substitutions and rearrangements but more indels.

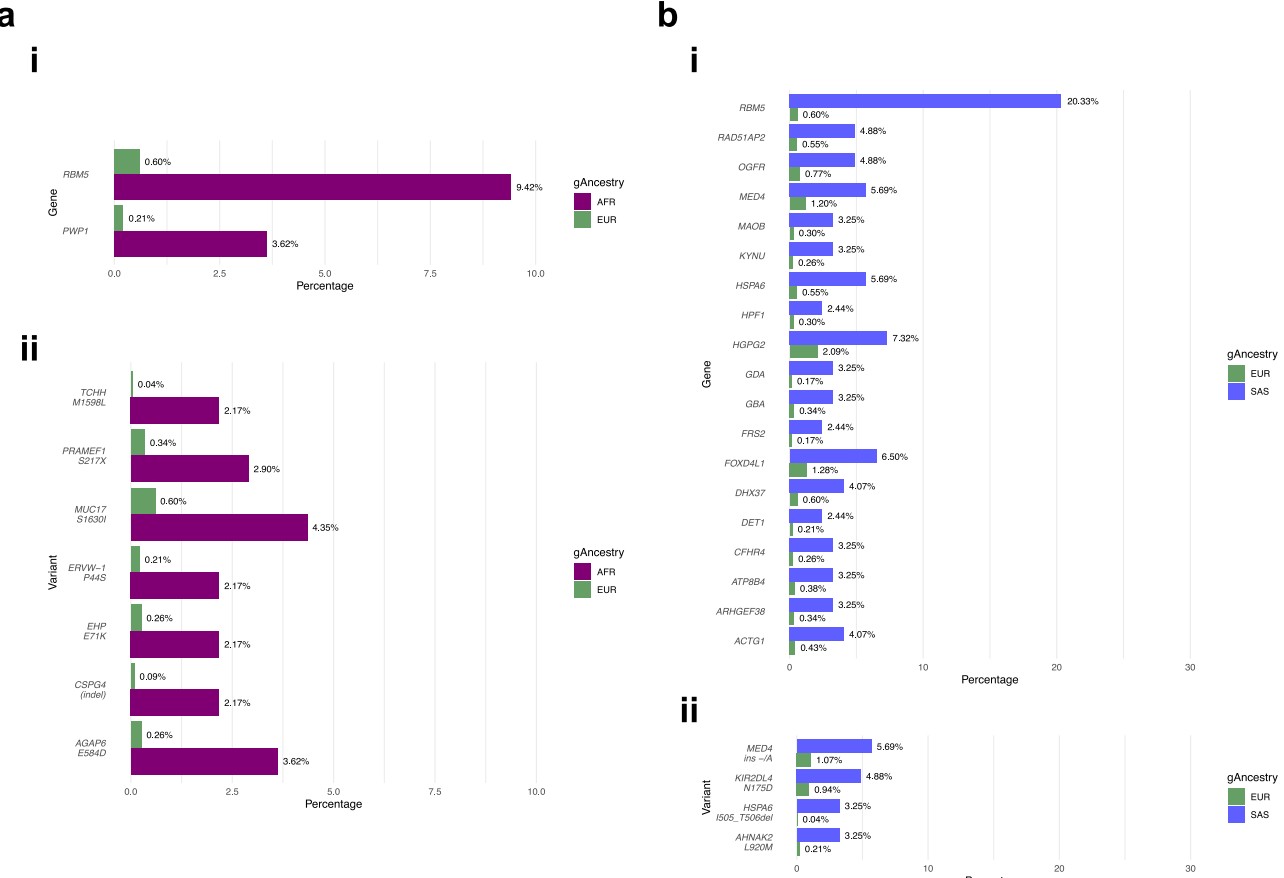

**Fig. 4 | Somatic genes and variants identified by the logistic regression model as differentially mutated in non-EUR versus EUR gAncestry in the Genomics England cohort. a** EUR v AFR comparison: ***i.*** significant genes; ***ii.*** significant variants; **b** EUR v SAS comparison: ***i.*** significant genes; ***ii.*** significant variants.

Three common signatures were identified as occurring in significantly differing proportions in the AFR or SAS group relative to EUR (Fig. 6a). SBS1 (associated with increasing age) and SBS3 (associated with homologous recombination deficiency, HR deficiency) were reported in high proportions across all gAncestry groups (SBS1 EUR 95.2%, AFR 99.3%, SAS 94.3%; SBS3 EUR 90.3%, AFR 97.8%, SAS 87.8%, Supplementary Data 9a), with a significant increase in frequency observed in the AFR population. SBS2 (APOBEC) was significantly lower in the SAS group relative to EUR. The rare SBS57 (identified as a possible sequencing artifact) was observed at a significantly higher frequency in the SAS group.

The double base substitution signature DBS12 (correlated with SBS105) was more prevalent in the AFR group (Supplementary Data 9b), with the other double base signatures showing similar proportions across all three groups. Rearrangement signatures RefSig R4 (dominated by clustered translocation patterns, associated with *CDK4* driver mutations) and RefSig R5 (characterised by unclustered deletions <100 kb) were more prevalent in AFR group, with a trend towards lower prevalence of both RefSig R4 and RefSig R5 in the SAS population (Supplementary Data 9c).

The supervised lasso logistic regression model HRDetect was used to determine HR-deficient tumours (Fig. 6b). There is a trend for more HR deficiency within the AFR population (17/138−12.3%) and lower HR deficiency within the SAS population (10/123−8.1%) against the EUR population baseline (225/2343−9.6%) (Supplementary Data 10a). While the AFR population exhibits higher proportions of SBS3, this trend is reversed in the stratified HR-deficient groups, with SBS3 presentation lower in the AFR HR-deficient group relative to that of the EUR population (Fig. 6). Similarly, the RefSig R5 rearrangement

contribution is also significantly higher in the AFR HR-deficient group. This suggests that the mechanism of HR deficiency could be different for different gAncestry groups. In HR-deficient SAS tumours, the contribution from deletions with microhomology is significantly higher compared to the HR-deficient EUR tumour baseline. Trends towards lower contributions from deletions with microhomology and loss-of-heterozygosity are seen in HR-deficient AFR tumours, although this is not significant. HR-deficient SAS tumours tended to have lower RefSig R5 contributions, but a similar loss-of-heterozygosity score to those of HR-deficient EUR tumours (Supplementary Data 10b).

Survival analyses[30] splitting the cohort into HR-deficient and HR-proficient groups showed that, although the mechanisms of HR deficiency are different between the cohorts, overall survival for HR-deficient tumours was significantly lower than HR-proficient tumours (Cox proportional hazard test $p = 8.49 \times 10^{-8}$, HR 2.4, Supplementary Fig. 6). However, when further stratifying the cohort by gAncestry, only the EUR remained significant (Cox proportional hazard test $p = 8.23 \times 10^{-8}$, HR 2.5).

## Pharmacological potential of genomic differences observed between the gAncestry groups

Eight significantly mutated somatic genes (*GPR158, FBXW7, BRIP1, CDKN2A, CHEK2, KDM6A, RET, STK11*) in the non-EUR populations were identified as candidates for therapeutic targeting in OncoKB (accessed 10/10/2024)[31], within clinical trials[32] and the literature (Fig. 7a)[33,34].

HR deficiency has also been associated with response to PARPi as well as a propensity for resistance to taxane therapies[35,36]. We assessed this on a specific cohort of BCN Biobank patients for whom complete longitudinal healthcare data were available from our local Barts NHS

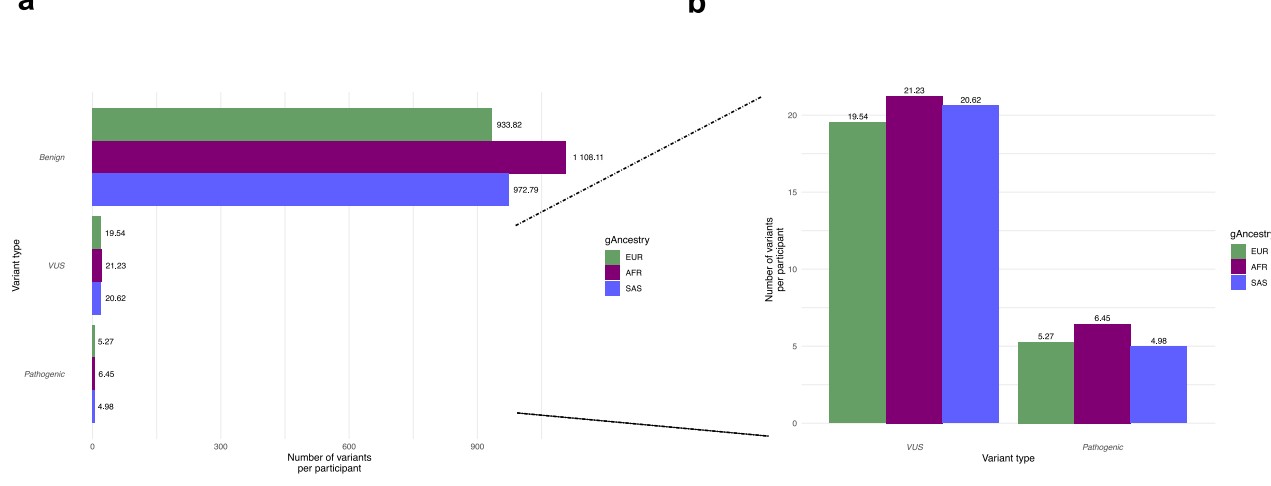

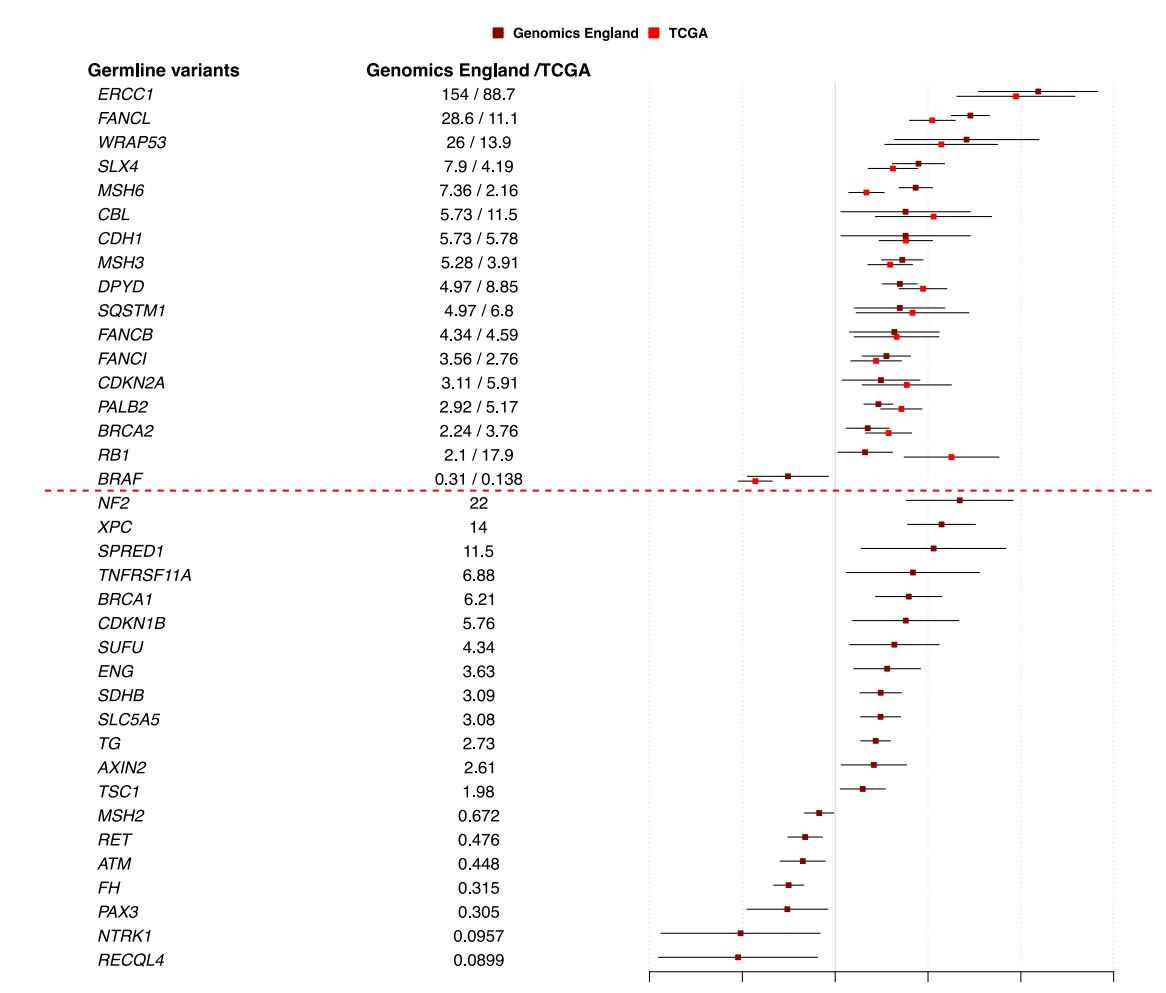

**Fig. 5 | Germline genes identified as differentially mutated by the logistic regression model in non-EUR versus EUR gAncestry in the Genomics England and TCGA cohorts. a** Pathogenic classification of germline variants across the gAncestry groups within Genomics England identify more pathogenic variants and VUS per patient in non-EUR populations. **b** Focus on the pathogenic and VUS variants per patient in non-EUR populations. **c** 45.9% (17/37) of the differentially mutated germline genes in Genomics England between AFR and EUR gAncestries were also differentially mutated in TCGA. ORs for Genomics England and TCGA shown (when difference is significant in either cohort), with 95% confidence intervals indicated from logistic regression modelling.

**Table 1 | Types of somatic variation present in each gAncestry, with comparisons made against EUR by two-sided Wilcoxon rank-sum test (\*<0.05, \*\*<0.01, \*\*\*<0.001)**

| | EUR (*n* = 2343) | AFR (*n* = 138) | SAS (*n* = 123) |
|---|---|---|---|
| **Substitutions** | 29,262,814 (12,489.5/patient) | 1,927,230 (13,965.4/patient) \* (*p* = 0.0161) | 1,398,052 (11,366.3/patient) (*p* = 0.868) |
| **Double substitutions** | 304,340 (129.9/patient) | 19,134 (138.7/patient) \*\* (*p* = 0.00939) | 15,733 (127.9/patient) (*p* = 0.275) |
| **Indels** | 13,733,867 (5,861.5/patient) | 660,062 (4,783.1/patient) \* (*p* = 0.0179) | 990,360 (8,051.7/patient) (*p* = 0.308) |
| **Rearrangements** | 436,604 (186.3/patient) | 31,063 (225.1/patient) \*\*\* (*p* = 3.75 × 10⁻⁴) | 21,672 (176.2/Patient) (*p* = 0.394) |

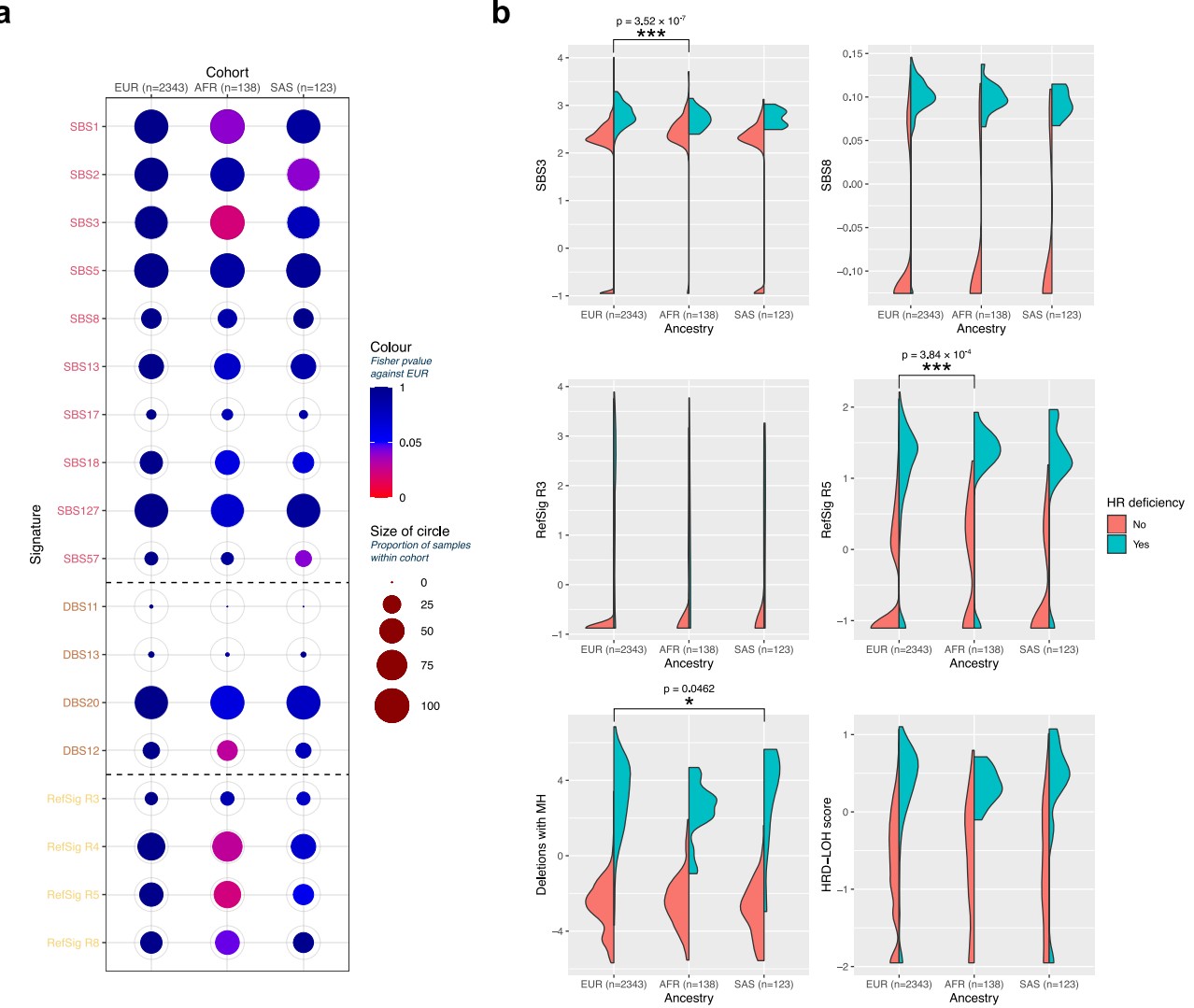

**Fig. 6 | Pattern of mutational signatures and homologous recombination deficiency between Genomics England gAncestry cohorts. a** Mutational signatures within each cohort with significance computed using EUR as reference. **b** Relative contributions of each component of the HRDetect scores used to determine *BRCA1*/*BRCA2*-deficient tumours in the gAncestry groups. Significant differences in each split violin compared to the EUR baseline (\*<0.05, \*\*\* <0.001, all *p*-values by two-sided Wilcoxon rank-sum test).

Trust and that best represented the EUR, AFR and SAS comparisons presented in this study (*n* = 160) (see Methods).

In this BCN Biobank pharmacogenomic exploratory cohort, we identified 17 (10.6%) patients as HR-deficient none of whom were administered PARPi. Furthermore, 70.6% of patients in the HR-deficient cohort were administered therapeutic regimens inclusive of a taxoid agent, of which 47.1% had a secondary event indicative of progression (e.g., recurrence, metastasis or death) compared to no events indicative of progression recorded in the non-taxane group (Fig. 7a).

Eight of the germline genes (*BRAF, BRCA1, BRCA2, DPYD, KIT, MET, PALB2, VHL*) tested were identified as enriched in non-EUR cohorts and as potential pharmacogenomic candidates (Fig. 7b). *BRCA1, BRCA2* and *PALB2*, known genes in BC gene testing panels, germline mutations have been shown to confer susceptibility to PARPi[37–40].

An eighth of participants in the BCN Biobank pharmacogenic cohort (12.5%) that were not administered PARPi as a part of their clinical journey had pathogenic germline mutations in *BRCA1/2* (Fig. 7b). Furthermore, when including *PALB2*, a gene mutated in significantly higher proportions in the AFR population relative to EUR, the

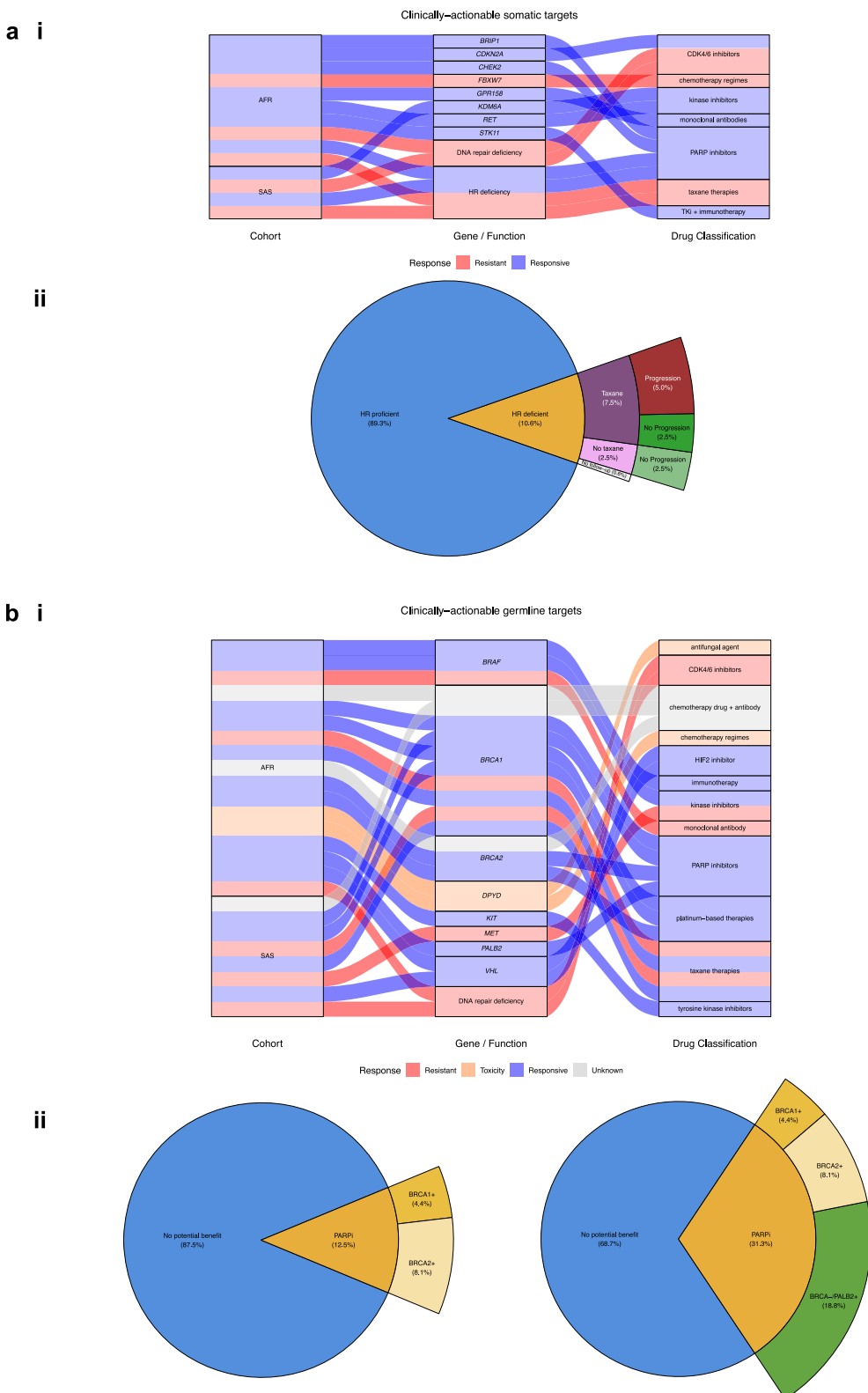

**Fig. 7 | Pharmacogenomic potential of the Genomics England cohort. a** Somatic variation. ***i.*** Alluvial plot of clinically-actionable targets. ***ii.*** Identification of therapeutic windows of opportunity within HR deficient populations and **b** Germline variation. ***i.*** Alluvial plot of clinically-actionable targets. ***ii.*** Identification of patients with (l) *BRCA1/2-* and (r) *PALB2-* mutations that could benefit from PARPi.

fraction of patients that may have benefitted from inclusion of PARPi as a part of the treatment regimen increased to 31.3% (Fig. 7b).

Survival analyses of the differentially mutated somatic and germline variants within our Genomics England cohorts split by gAncestry[30] is limited by the short length of follow-up data on several participants, leading to a dominant right-censored data effect that adds bias to the survival estimates, and so was not performed due to the small numbers in each group.

## Discussion

We present a comprehensive analysis of clinical and molecular features associated with African and South Asian gAncestry from a cohort of 7253 breast tumours. Our findings highlight the importance of addressing the racial disparity in research to optimise precision oncology, ameliorating outcome, and guiding the patient clinical journey.

Each cohort examined in this study uses different methods for calling germline variants (Methods). The callers applied display similar sensitivities for SNVs and short indels on tissue samples, but TCGA's approach of taking a consensus call increases confidence in their final variants[41]. Similarly, for somatic small variants, both Strelka2 (Genomics England) and Mutect2 (TCGA) use a statistical model for calling variants from paired samples but differ in their application of the model: Mutect2 uses a Bayesian classifier and Strelka2 uses a trained machine learning method. Mutect2 trades accuracy for lower sensitivity, but Strelka2 increases sensitivity at the risk of false positives, but on paired tissue samples, their results are comparable[42].

It is not recommended to use self-identified population descriptors as proxies for gAncestry groups due to their categorical and continuous properties, respectively[43]. So, while we report a high concordance between self-reported ethnicity or race and gAncestry, our study focusses on use of gAncestry as an analytical variable across all cohorts, bar those patients in the BCN Biobank for whom genomic data and associated gAncestry derivations are not available[43]. As such, SRE derived from primary and secondary EHRs were used to inspect overarching trends observed in analyses focussing on the gAncestry groups studied but were not used an analytical determinants per se.

Use of the latter for the stratification of populations not only promotes equity, clarity and reproducibility in research methods but also allows for greater examination into the genetic diversity of a population and how this diversity influences disease pathology and response to treatment[3,14,44].

When considering clinical associations whose effect sizes are biologically meaningful, we report non-EUR/non-White populations to present at younger ages, experience more clinically aggressive HR- disease and suffer from a higher mortality rate relative to the EUR/ White population[5,7,45,46]. For the Genomics England cohort, where IMD is available, the distribution of IMD within non-EUR participants within our dataset is skewed to higher deprivation quintiles compared to EUR participants. Testing IMD as a confounding factor did not reduce significance, aside from patients presenting on average 3 years younger when comparing within the EUR cohort. Despite this, it is important that patient cohorts are socio-economically matched as closely as possible to reduce this potentially confounding effect. Even after this matching, it has been shown that long-term activation of the stress response—the cumulative biological burden extracted on the body's systems due to repeated adaptation to stressors over time (allostatic load)—may also significantly influence disparities in presentation, progression, and treatment[47]. Similarly, systemic barriers to accessing health care also contributes to differing outcomes[48,49].

It is debatable as to whether current screening guidelines, in which a single age-based window is applied, benefit all women equally[5,50–54]. The AgeX and UK Age trials examined the feasibility of lowering the age at which patients are first invited for routine BC screening. While our study did not examine mortality statistics, results from the AgeX trial, which aimed to amend the screening window to 47–73 years, reported a 24% reduction in mortality[51,52]. Similarly, the UK Age trial, which investigated the benefits of initiating mammography screening at the age of 40 or 41, demonstrated a 25% reduction in BC mortality after 10 years[28]. However, these findings did not examine ethnicity as an independent factor, thus the long-term benefits of these strategies on reducing BC mortality in women from ethnic minority groups remain largely unknown.

Our findings support the premise that current screening guidelines detect imbalanced proportions of BC between the gAncestry groups and that ethnicity–adapted screening windows would allow for greater equity in the screening process. This includes adjusting the screening range for non-EUR cohorts to start at a younger age. In the Genomics England cohort, these revised windows would be EUR 50–75 years (to detect 59.47% of BCs), AFR 47–69 years (to detect 61.31% of BCs) and SAS 45-68 years (to detect 59.35% of BCs). Applying these screening windows to BCN Biobank patients not dually-consented with Genomics England gives increased coverage—White 67.4% (793/1176), Black 64.8% (190/293) and Asian 70.5% (148/210)—as does applying these new windows to the G&H data improving from 26.4% (96/363) to 59.2% (215/363) within this group. The extra coverage in BCN Biobank patients could be explained by the potential effect of the socio-economic factors (as represented by estimated IMD) for these groups compared to Genomics England.

The appropriate designation of a bespoke screening model, which would provide an indicative screening window and guide the modality to be used during the screen, requires the interplay of multiple complex factors, such as demographics, family history of inheritable cancers, mammographic density, previous benign breast conditions and the results of genetic testing for gAncestry-based susceptibility genes. Input from patient advocates and representation from social groups with low current uptake are also necessary to increase the penetrance of earlier intervention, working to reduce reluctance in seeking medical assistance or taking part in screening.

Literature associating TMB with gAncestry or ethnicity reports that AFR patients tend to have a higher TMB relative to EUR[55,56]. The corresponding trend in Asian patients is equivocal, with reports of TMB being both higher and lower in this group relative to EUR[55–57]. This is likely due to the combination of the presence of EAS and SAS gAncestry within Asian study groups, with no distinction made between the two.

We found TMB to be significantly lower in the SAS population, with a similar trend in the AFR population, compared to that in the EUR group. When patients are stratified based on screening age (50 years), TMB is not significantly different between non-EUR populations and their corresponding EUR age group, but there is a trend towards higher TMB in the <50 years AFR subgroup. These observations could be due to the association of increased TMB in ER- breast cancer, of which a higher proportion are present in the <50 AFR subgroup compared to the EUR counterpart.

Although, overall, higher TMB is associated with poorer survival in many cancers including BC[58], very high TMB ($\geq 50$ mut/Mb) has been reported to have a protective effect in immunotherapy-naïve patients[59], where it is attributed to increased cell lethality from extreme genetic instability. TMB, both in isolation and in combination with expression markers, has also been reported as a potential predictive biomarker of response to immune checkpoint inhibitors (ICIs) and has been associated with better treatment response and improved outcomes[60,61].

Estimating TMB from tumour-only sequencing (whether WGS, WES or targeted gene panels) is commonly used to determine suitability for ICI treatment, but this may exacerbate disparities between ethnic groups. This is because reference data is predominantly based on EUR gAncestry groups, meaning non-EUR gAncestries are likely under-represented. This reference data is used to identify and remove potential germline variants and true non-EUR gAncestry germline variants can persist after filtering, inflating the estimation of TMB in ethnic minority groups and the use of ICI treatments. Correcting the estimated TMB for gAncestry in this case is imperative[62]. The present study used somatic variants called from paired sequencing, allowing the patients' own germlines to be used for accurate calculation of the TMB.

Six genes with somatic variants differentially mutated between EUR and AFR in the Genomics England cohort (*RBM5, OTOF, FBXW7, NCKAP5, NOTCH3, GPR158*) were also identified in the TCGA for the same comparison. These genes have been associated previously with key roles in tumourigenesis and progression[6,63,64], and as diagnostic predictors poor prognosis in other solid tumours[65–67]. Mutations in *FBXW7* are found across multiple cancer types in AFR populations[6], suggesting this alteration is not limited to BC. Mutations in *FBXW7* and *GPR158* have been identified as therapeutic candidates and have been associated with sensitivity to lunresertib and camonsertib in ovarian cancer[68,69], sensitivity to bevacizumab in osteosarcoma[34] and resistance to paclitaxel in BC[70].

Although both the AFR and SAS cohorts presented with higher grade tumours compared to EUR, none of the somatic differentially-mutated genes were found in the literature have any association to the aggressiveness of the disease. Higher grade tumours are associated with more ER-/HER2- cancers, and also with germline mutations in DNA damage repair genes, which both these cohorts exhibit higher proportions of when compared to the EUR cohort.

In addition, six other somatic genes significantly enriched in non-EUR populations in the Genomics England cohort (*BRIP1, CDKN2A, CHEK2, KDM6A, RET, STK11*) were also identified as pharmacogenomic candidates. Mutations in these genes have been reported to confer sensitivity to PARPi[38,40], CDK4/6 inhibitors[71], EZH2 inhibitor[72], RET inhibitors[73] and combination tyrosine kinase inhibitors with immunotherapy[74], respectively, in studies on BC as well as other solid tumours. These findings, suggest repurposing opportunities of some of these drugs to treat breast tumours in non-EUR ancestries.

Mutation signatures and HR deficiency inferred from somatic genomic alterations (SNVs/indels and CNVs/SVs) show differences between the three gAncestry Genomics England cohorts. SBS signature 8, associated with HR deficiency, and SBS signatures 2 and 13, related to *APOBEC3A* and *APOBEC3B* function, are enriched in AFR (SBS2 and 13) and SAS (SBS8) populations, and have also been reported enriched previously in non-EUR BC[75]. There is a trend towards more HR deficiency in the AFR population and less HR deficiency in the SAS population compared to the EUR population baseline. Splitting the cohort into HR-deficient and HR-proficient tumours shows that overall survival for HR-deficient tumours is significantly lower than HR-proficient tumours, although following stratification by gAncestry only the EUR remained significant. This is attributable to two factors: the small numbers of HR-deficient patients with survival data within the non-EUR populations, leading to large confidence intervals; and that other clinical features may be colliding with HR deficiency and influencing outcome. For instance, our analysis may exhibit bias as patients in non-EUR gAncestry populations tended to present with higher tumour grade (potentially due to increased ER- and/or enrichment of *BRCA1/2* mutations), which is associated with decreased overall survival. With larger cohorts, allowing for more stratification on tumour grade and HRD status, we would be able to determine how much of the effect on survival was due to HR deficiency, and how much was due to presentation at higher grades in non-EUR participants.

HR deficiency classifications inferred from whole genome sequencing attempt to classify *BRCA1/2* deficiency and identify those patients that exhibit mutational landscapes similar to those with *BRCA1/2* defects i.e., BRCAness. The HR deficiency findings support our germline observations in which a higher proportion of germline *BRCA1, BRCA2* and *PALB2* mutations are present within AFR and SAS populations. Identification of these genomic alterations provides opportunities by highlighting potential therapeutic vulnerabilities and could be exploited for patient benefit. For instance, research has reported on the clinical efficacy of PARPi, either as a single agent or in combination with existing treatment regimens, in ameliorating outcome in BC patients with germline mutations in *BRCA1/2* and *PALB2*, as well as BRCAness[38–40]. Similarly, *BRCA1/2*-mutated tumours have also

been reported to respond better to platinum therapies[76]. However, BRCAness is known to confer resistance to taxane-based chemotherapies[77].

We identified a subset of patients in our BCN Biobank pharmacogenomic exploratory cohort with drug responses associated with specific germline mutations and/or somatic mutation patterns associated with HR deficiency. Almost 40% of these patients presented with a potential sensitivity to PARPi to whom this was not administered as a part of their therapeutic regime(s). Similarly, we also identified a subset of patients potentially resistant to common therapies in the HR-deficient group.

Our findings also show that the landscape of BC-associated susceptibility genes differs between the gAncestry groups. With current clinical genetic panels developed from research that would have been ethnically biased towards a White population, the panel may not accurately represent the mutations, frequency of mutations or risk of cancer in ethnic minority populations. In agreement with previous studies, our findings show that the landscape of BC-associated susceptibility genes differs between the gAncestry groups, suggesting that germline screening protocols modified based on ethnicity could be more informative[11,78,79].

The penetrance of *BRCA1* and *BRCA2* is known to be modified by SNPs that influence risk in general population, but the landscape of germline diversity within African populations has yet to be explored, and so there are potentially other ancestry-related germline risk modifiers that are yet to be found[80]. A similar issue arises for germline diversity within Asian populations, with SAS and EAS gAncestries each having their own distinct *BRCA1/2* mutation patterns, which are comparatively less examined[78,81].

Germline mutations in DNA damage repair genes, such as *BRCA1, BRCA2, CHEK2* and *ATM*, are associated with lower efficacy of CDK4/6 inhibitors and endocrine therapy in advanced BC[82,83]: This is particularly relevant within the non-EUR populations that tend to present at a later stage.

The clinical implications of the benefits of PARPi and CDK4/6 inhibitors discussed above are not specific to a single ethnicity but, instead, are of benefit to all patients. As such, increased genetic testing of patients, using routinely genomic tests from BC clinics and the somatic profiling of tumours could guide the personalised therapeutic management of disease.

Our findings highlight the broader issue of under-representation of ethnic minority groups in research and the historical merging of East Asian and South Asian populations biasing the translational implications and resources developed as a result. Use of references developed from research based on predominantly White populations to categorise mutations and define pharmacogenomic implications has been shown to have broader implications in terms of ethnicity-associated response to treatment. For instance, it has been shown that variations in *BRCA1/2* in resources such as ClinVar, are not representative of global ethnic populations[78]. Another study based on the G&H cohort identified ethnicity-driven pharmacogenomic consequences unique to British South Asian patients with cardiometabolic disease. Common polymorphisms in *CYP2C19* in this population have been associated with poor activation of clopidogrel and recurrent myocardial infarction[84].

We report an imbalance in our research data from Genomics England, TCGA and BCN Biobank. These three data sources are not representative of the countries they derive from, due to the limited number of collection sites in each. The largest estimate of the South Asian American population in the US is 2% of the total US population (US Census Bureau estimate, 2021), which is exemplified by the low percentage of SAS patient data available not only from TCGA but also from the ICGC Pan-Cancer Analysis of Whole Genomes project (PCAWG)[6]. This reduces its utility as validation of our results for the SAS gAncestry group if the population is sampled randomly. Thus,

while the TCGA dataset can only be used to validate the molecular and clinical data differences between EUR and AFR ancestries, the BCN Biobank clinical data can be used to validate clinical features across all three of our gAncestry cohorts, notwithstanding that the lead collection site of BCN Biobank is in North East London, where the IMD distribution is skewed to more deprived quintiles.

The paucity of data on a group that comprises up to 20% of the world's population is now recognised[85], with initiatives attempting to reduce this research inequality. Our cohort has the largest proportion of SAS ancestry patients among studies of similar size, reflecting the population structure of England (Supplementary Fig. 2a). In fact, METABRIC[86], examining 2000 tumours, focused more on molecular subtypes and did not report ethnicity or gAncestry differences, leaving our study as the largest study which specifically examines these. The under-representation of the SAS ancestry group in international cancer-associated studies does, however, mean that the variants we report are difficult to validate in publicly available datasets. The scarcity of the stratification of Asian ethnicities in cancer research means that any study of genomic variation using these merged population data would likely be underpowered to identify alterations unique to the South Asian population. In fact, data from large-scale US-led initiatives are likely biased towards East Asian ethnicities (and ancestries) due to them recapitulating their sampling populations.

Our genomic findings are limited by the relatively small sizes of the AFR and SAS groups (120–140 patients) within our Genomics England analytic cohort of 2781 patients. However, this is still larger than most studies which aim to determine gAncestry-related genomic differences within the BC landscape. Nevertheless, this could be improved through the inclusion of more somatic tissue sequencing from our other British-based cohorts, but the overall ratio (of EUR to AFR or SAS patients) would continue to exhibit bias towards the EUR gAncestry, unless the non-EUR groups within the UK (or worldwide) were specifically targeted for collection. However, our study was still powerful enough to find significant differences between EUR and the other two groups. Any further study would need to specifically target the non-EUR groups either within the UK or worldwide to move towards equipoise and extract the more subtle variations that may not currently be apparent.

Tackling disparities in research has become a public health priority. One of the Cancer Grand Challenges[87] is focused on cancer inequities; Genomics England has recently implemented the Diverse Data initiative[85] to bridge the ethnic data gap in genomics-driven personalised medicine; and the BCN Biobank is focusing on increasing recruitment from ethnic minority populations. Furthermore, the Breast Cancer Now's Inequalities Funding Scheme was implemented to encourage applications for research into increasing health equity within BC. These shifts in future data and sample collections will help ensure that findings from research will be more powerful due to better equipoise in ethnic groups ensuring equitability in translational implications.

There are evident health disparities in BC diagnosis, therapeutic management, and outcome globally. While determining genetic diversity is important for the advancement of precision oncology, disparities in healthcare cannot be attributed to a single factor but rather stem from a complex web of interlinking clinical, social and genetic factors. To ensure that precision oncology benefits all patients equally, regardless of their ethnic background, it is imperative to foster trans-disciplinary co-operation and conduct multi-modal studies, that incorporate data from primary and secondary healthcare in addition to genomics.

## Methods

### Ethics statement

East of England - Cambridge Central Research Ethics Committee gave ethical approval for the Breast Cancer Now Biobank (REC reference 23-

EE-0229). Genomics England Clinical Interpretation Partnership (GECIP) gave approval for this work (reference 643) to use Genomics England data. Genes & Health gave approval to this work reference S00087 to use the Genes & Health data. The Data Access Committee for The Cancer Genome Atlas (TCGA) gave approval for this work reference dbGaP project #15970 to use TCGA data.

All necessary patient/participant consent was obtained and the appropriate institutional forms archived, and that any patient/participant/sample identifiers included were not known to anyone (e.g., hospital staff, patients or participants themselves) outside the research group and so cannot be used to identify individuals.

### Data collection and collation

**Genomics England clinical data.** For the clinical data for our Genomics England cohort, the following tables within the Genomics England TRE (v17; March 2023) were queried using LabKey via the R/LabKey API. The relevant participant IDs were identified from the *cancer_participant_disease* table, with other primary and secondary tables queried using these unique Participant IDs as primary keys (Supplementary Methods). This gave an initial BC Genomics England cohort of 3336 participants of female sex as determined through karyotyping of the germline data.

One participant was in the database twice under two participant IDs, flagged under a *duplicate_participant_id* entry; one further participant withdrew consent before the v17 data release, leaving 3334 BC participants in the initial investigative cohort.

For each selected data item, a single entry was consolidated for each participant, using the primary data source. In the absence of primary data, data from the secondary source was used, with information extracted from the record closest to diagnosis date, unless explicitly stated otherwise (Supplementary Methods). Upon collation, the clinical information of the patient with duplicated IDs was assessed for discrepancies and the duplicated information was removed, with the earliest date selected to represent date of diagnosis.

The final analytical cohort was determined following clinical and genomic criteria for exclusion as described in the previous section (Fig. 2), with three further samples removed following QC checks based on TMB calculations.

**BCN Biobank clinical data.** The BCN Biobank extract (accessed 27/04/2023) comprised 2479 patients (of self-reported female gender) from Barts Health (BH) NHS Trust. BH is the largest NHS trust in London with five hospitals (St Bartholomew's Hospital, The Royal London Hospital, Mile End Hospital, Newham Hospital and Whipps Cross University Hospital) serving 2.5 million people across the diverse population of North East London. BCN Biobank-Barts data are linked to EHRs from BH via the NHS or hospital number. 2126 were female and had complete clinical data associated with incidence of unilateral primary BC. Of these, the genomic data from 195 samples were available from the Genomics England TRE as these patients were consented by both BCN Biobank and Genomics England. Basic demographic statistics (age, ethnic group) was updated using linkage between this dataset and the primary care data set from Discovery East London programme (extract date 16/11/2023)[26]. Concordance between reported ethnicity (within BCN Biobank) and gAncestry for the dually-consented patients is 88.5%, for the primary care data alone the concordance is 86.8%. Using primary care ethnicity data when not specified in BCN Biobank improves concordance between the reported ethnicity and gAncestry to 94.5%. These dually-consented participants (BCN Biobank and Genomics England) form the basis of our BCN Biobank pharmacogenomic exploratory cohort.

With the South Asian ethnic group being a focus of the study, participants from "Any Other Asian Background" were moved to the "Other" ethnic category to prevent dilution with possible Middle Eastern, Chinese or other South-East Asian ethnic groups.

The calculated BCN Biobank age at diagnosis was used within the Genomics England cohort for those patients who were dually consented.

**TCGA clinical data.** TCGA clinical data for the TCGA-BRCA study ($n = 1098$) was downloaded from the Genomic Data Commons Data Portal[88], and filtered for all those cases with neoplastic breast disease ($n = 1076$). This was further filtered to retain self-reported female patients, leaving 1064 cases.

**G&H clinical data.** We used the G&H clinical dataset (December 2023 release), selecting 57,047 volunteers with basic demographic information available (55.4% self-reported as female; 44.6% self-reported as male) through primary care records from Discovery East London programme or secondary care records from Barts Health NHS Trust and NHS Digital. We filtered the available data to define individuals with BC ($n = 364$), if the EHR contains a clinical code (ICD-10 or SNOMED CT diagnosis) indicative of the present or past malignancy of breast.

**Genomics England genomic data.** All genomic data is accessible from the Genomics England TRE. Samples of germline genetic material and tumour tissue were sequenced, mapped to the hg38 genome assembly, and variants called: germline variants using the Isaac single sample short variant caller and somatic short variants using Strelka2[89] on the somatic and matched germline. Larger copy number changes and structural variants were called using the Manta (structural variant calling)[90] and Canvas (copy number changes)[91] pipelines. Genomics England reports of small somatic variants (SNVs and indels) were obtained from the *cancer_tier_and_domain_variants* table, with locations for the raw germline vcf data held in the *cancer_analysis* table.

Ancestry inference was conducted by Genomics England as described previously[92]. In brief, a random forest classifier was applied on the first 8 principal components derived from 188,382 good quality SNP locations within the 1000G phase3 data. The classifier generated 400 trees, with the proportion of trees classifying a germline vcf into a gAncestry superpopulation (AFR, EUR, EAS, SAS, AMR). Assignment of a sample into a gAncestry group was performed using a $\geq 0.8$ proportion threshold, with individuals in the Admix group defined as those without a gAncestry proportion exceeding the defined threshold. The superpopulation proportions for each vcf are available from the *aggregate_gvcf_sample_stats* table in Genomics England.

Up-to-date information from EHRs, comprising structured and unstructured data, was accessible for the patients dually consented by both Genomics England and the BCN Biobank, with clinical data from other resources showing propensity for being relatively sparse and/or fragmented. A BCN Biobank pharmacogenomic exploratory cohort ($n = 160$) was compiled from this dually-consented group, taken from those participants which were in the Genomics England EUR, AFR and SAS gAncestry cohorts.

**TCGA genomic data.** Simple somatic variation data for all TCGA-BRCA patients was downloaded from the GDC data commons (under dbGaP project #15970) and then filtered using the TCGA participant ID for our TCGA cohort. This file arises from paired-sample consensus calling using MuSE[93], VarScan2[94], Pindel[95] and Mutect2 from GATK[96]. The gAncestry calls were accessed from a previous study[6] that defined the consensus ancestry call from five calling methods.

Germline data for the TCGA-BRCA cohort was downloaded from the PanCanAtlas[97] from the analysis of 10,389 TCGA cancers and filtered for those participants which overlap our TCGA analyses cohorts. This file is based on consensus variants arising from multiple variant calling methods (MuSE[93], VarScan2[94], Pindel[95] and HaplotypeCaller from GATK[96]).

**G&H genomic data.** Germline data for the G&H cohort was available within its Trusted Research Environment for 44k participants (July 2021). This data release, arising from GATK HaplotypeCaller with hard post-calling filtering on WES data was used to derive for our breast cancer cases ($n = 296$) and gAncestry-matched control ($n = 17,626$) participants.

### Data analysis

**Demographic and clinical features.** Demographic data from the non-White/non-EUR cohorts were compared against the White/EUR reference using methods described previously[6]. Where age at diagnosis and age at death were available, a linear model was applied; similarly, a scaled negative binomial model was applied to the tumour-mutation burden. For other variables, a logistic regression was performed, with upper levels of the factor compared against the reference level. For Genomics England, upper levels with small numbers (particularly in the AFR and SAS gAncestry groups) were grouped together. For receptor statuses, positive receptor status was taken as the reference level. In all clinical comparisons, for the Genomics England cohort, the gAncestry groups were jointly modelled using unordered gAncestry as a factor. To confirm the effect sizes in a direct AFR against SAS comparison, we split these two groups and ran the regressions separately.

In Genomics England, because of the potential confounding effect of IMD on demographic data, we performed two tests. Firstly, a nested likelihood ratio test of the demographic factors was conducted, comparing a reduced model (gAncestry as sole predictor) with a full model (gAncestry and IMD as predictor variables), using the lmtest R package (v 0.9-40). Secondly, the association between IMD and other demographic factors were tested within the Genomics England EUR gAncestry group, using the arsenal R package (v 3.6.3).

**Tumour mutational burden.** Tumour mutational burden (TMB) for the Genomics England cohort was calculated following the method previously described[62], correcting for gAncestry. The chosen gAncestry reference population within gnomAD[98] was AFR for AFR, SAS for SAS and NFE (non-Finnish European) for EUR. Non-synonymous somatic variants within exonic regions were filtered based on the relevant gnomAD population frequency, the COSMIC database[99] and the TOPMED (v3) database[100].

A mutation was retained if it satisfied one of the following criteria: (i) gnomAD population frequency was $\leq 0.1\%$ (rare) and the VAF was $\geq 3\%$ or if the gnomAD population frequency $\leq 0.1\%$ and the VAF was $< 3\%$ and annotated by at least two COSMIC identifiers or (ii) if the relevant gnomAD population frequency was $> 0.1\%$ but $\leq 10$ and annotated by at least two COSMIC identifiers (Fig. 8). The variant was discarded if it appeared in the TOPMED database at a frequency $> 0.1\%$. The count of retained mutations per sample was divided by the size of the exome ($\approx 35.4$ Mb) and multiplied by $10^6$ to calculate the TMB per Mb of exome.

**Differential somatic variation determination (Genomics England cohort).** Once the filtered variant list from the TMB calculation was determined, both mutated genes and the individual variants were collated separately. For each gene or variant in the lists, a logistic regression was performed of the following form (comparing the three gAncestry groups jointly as an unordered factor):

$$Gene \text{ or } variant \sim gAncestry + Age \text{ at diagnosis}, \qquad (1)$$

where the gAncestry reference level was EUR. If the adjusted p-value for the gAncestry odds ratio was less than 0.1, the gene or variant was considered differentially present[6]. However, in the cases where variants were not present in one of the two comparative cohorts, the algorithm did not converge, and so a second method was employed.

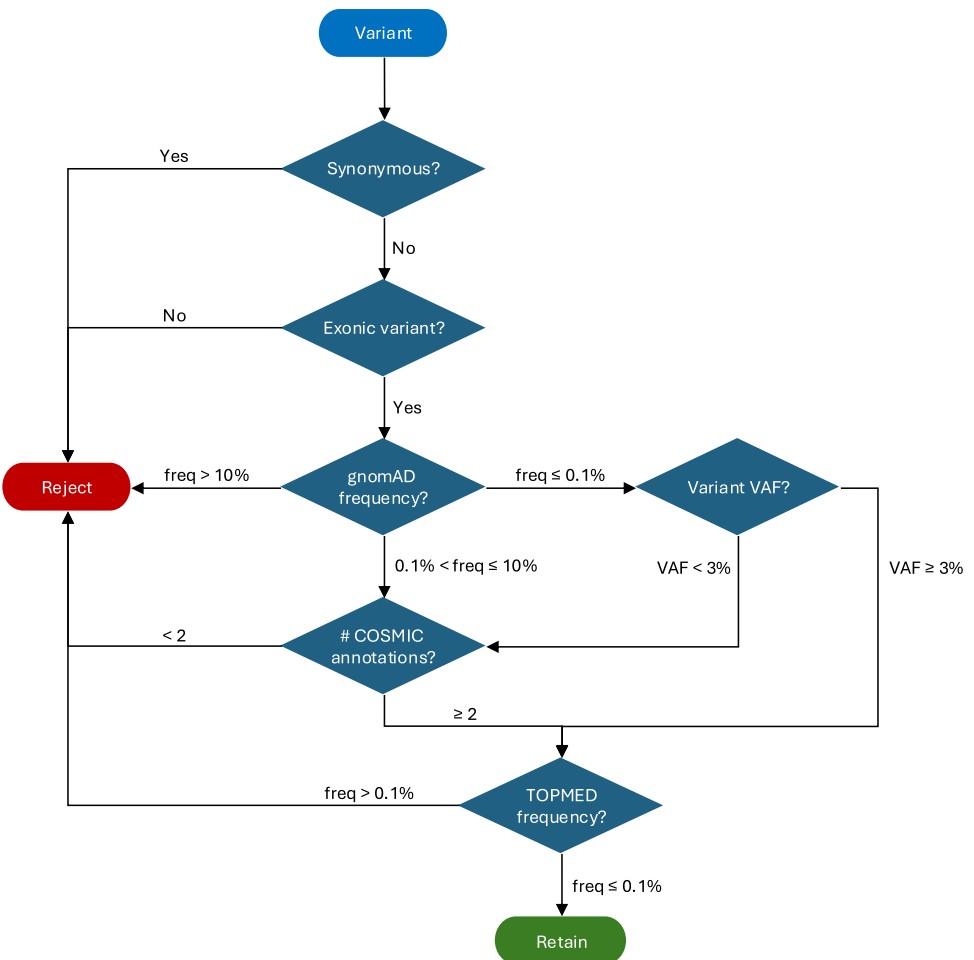

**Fig. 8 | Method for determining retained somatic variants for TMB calculation.** Somatic variants are retained if non-synonymous and in exonic regions, with other inclusion/exclusion criteria dependent on the external databases gnomAD, COSMIC and TOPMED (v3) as detailed in the text (see Methods).

In this second method, mutated genes or variants were marked as present or absent in a cohort if present at above or below 2% in the gAncestry group (a threshold of 47/2343 in EUR, 3/138 in AFR and 3/123 in SAS). The thresholded lists of genes and variants present in each cohort were intersected to give seven sets of intersections (EUR only, AFR only, SAS only, EUR + AFR, EUR + SAS, AFR + SAS, EUR + AFR + SAS). For our purposes, we examined those genes or variants present at above 2% in AFR only and above 2% in SAS only, with the implication that they are present at <2% in the other two cohorts.

To control for the potential confounding factors of germline *BRCA1* and *BRCA2* status and IMD in the logistic model, we added extra terms and restricted to those participants for which all the data was available:

$$Gene \text{ or } variant \sim gAncestry + Age \text{ at diagnosis} + BRCA \text{ status} + IMD. \tag{2}$$

For the logistic regression model, given the sample sizes for each gAncestry within Genomics England, we have 90% power to detect an OR 5.86 (SAS)/5.12 (AFR) down to 1.97 (SAS)/1.89 (AFR) as mutation frequency increases from 5% to 25% in the minority gAncestry.

**Germline variations (Genomics England cohort, TCGA and G&H).** The genes we selected for analysis were combined from a list of those tested in the clinic (as reported in the NHS Genomic Medicine Service Panels v8[27]) and those conferring susceptibility to cancer (as reported in the Genomics England PanelApp gene panel[28]), leaving 180 genes in total.

For Genomics England and TCGA, germline vcf files were passed through a pipeline to collate and annotate variants in selected genes using bcftools and VEP (version 107)[101]. For determining pathogenicity, we restricted to the canonical transcript and filtered further using the pathogenicity classification and confidence from ClinVar (Fig. 9).

For G&H, the joint germline WES vcf, which had been previously annotated using VEP version 107, was filtered on the genes and the participants in our case and control and passed through the selection criteria (Fig. 9). The G&H WES release did not have variants on chromosome X because of ploidy issues when using GATK HaplotypeCaller, so genes from our list located on this chromosome were not included for the case:control study.

As per the demographic calculation, a logistic model was fitted for the forest plots (with EUR in Genomics England/TCGA or control as the reference level for the *factor*):

$$variant \sim factor. \tag{3}$$

Note that in the Genomics England cohort, the factor is the unordered gAncestry with EUR as the reference level.

**Signature and HRDetect calculation (Genomics England cohort).** The HRDetect[29] predictor was used to detect homologous recombination-deficient tumours within our Genomics England analytical cohort. Here, the somatic short-variant vcfs available within the Genomics England TRE were split into SNVs and indels without filtering and the somatic copy number vcfs were split into SVs and CNVs. The R

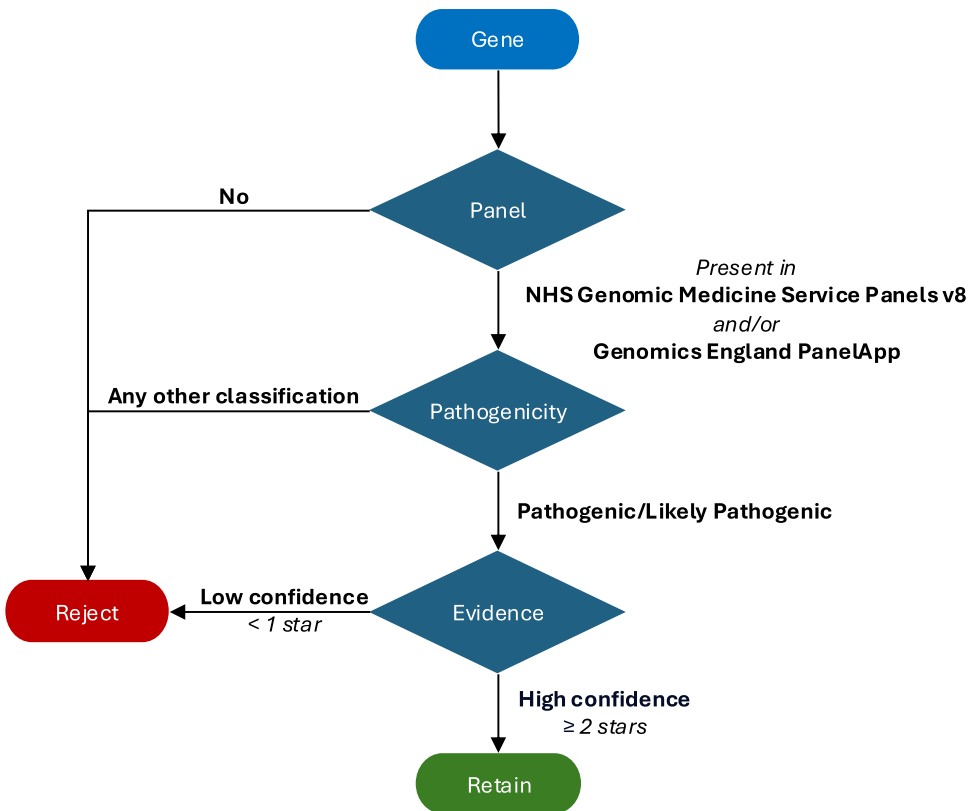

**Fig. 9 | Method for determining germline pathogenic/LoF mutations.** Germline variants are retained if the gene is present in the NHS Genomic Medicine Panels v8 or Genomics England PanelApp (or both), and are annotated as pathogenic/likely pathogenic by ClinVar with high confidence of 2 stars or higher.

library was used to process the SNVs into SNV and DNV catalogues, and the SVs into SV catalogues. Finally, the HRDetect function was applied on the four categories of somatic variants (SNPs, indels, SVs and CNVs) to compute the homologous recombination deficiency score using their pretrained classifier.

**Survival analyses (Genomics England cohort).** Survival analysis was performed as in the recent pan-cancer Genomics England paper[30], using the code and methods provided, on our cohort, both as a whole, and split between the three gAncestry groups EUR, AFR and SAS.

**Reporting summary**

Further information on research design is available in the Nature Portfolio Reporting Summary linked to this article.

## Data availability

Genomic and phenotypic data for the 100K GP study participants are available through the Genomics England Research Environment via application at https://www.genomicsengland.co.uk/join-us: approval to access the anonymised data through the Genomics England Trusted Research Environment requires a research project proposal, and mandatory training on information governance. The clinical data and donor genetic ancestry calls for the TCGA cohort used in this study are available from the supplemental information (Table S1) of Carrot-Zhang et al.[6] (https://doi.org/10.1016/j.ccell.2020.04.012; Table S1 at https://ars.els-cdn.com/content/image/1-s2.0-S1535610820302117-mmc2.xlsx) Additional clinical and genomic data for the TCGA BRCA cohort are accessible from the Genomics Data Commons Data Portal (https://portal.gdc.cancer.gov/), with the clinical data openly downloadable; somatic data are downloadable following application to dbGaP (https://gdc.cancer.gov/access-data/obtaining-access-controlled-data,

accession ID phs00178, [https://www.ncbi.nlm.nih.gov/projects/gap/cgi-bin/study.cgi?study_id=phs000178.v11.p8] for the controlled somatic variation data[88]. Germline TCGA-BRCA data is available from the PanCanAtlas[97], again through applying through dbGaP for the controlled data in the GDC portal under the same accession ID (phs00178). Clinical data for BCN Biobank participants is available (anonymised) on application to the Biobank (https://breastcancernow.org/our-research/information-for-researchers/apply-to-our-biobank/how-to-apply-to-the-biobank, https://biobank.bcc.qmul.ac.uk/surveys/?s=KEAEJ4RFRH). Clinical and molecular data from G&H is available through the Genes and Health Research Environment via application at https://www.genesandhealth.org/researchers/apply-for-access/: applicants are required to fill in a Data Access Agreement, complete mandatory information governance training and pay a small fee for access to the trusted research environment to examine individual-level data. For the restricted data classes, summarised data can be exported from the trusted research environments for research purposes with restrictions to prevent the identification of individual participants through data linkage.

## Code availability

Codes for all the analyses from the manuscript will be made available to researchers on application to the authors via a release from a GitHub repository, under a Creative Commons Attribution-NonCommercial 4.0 licence (CC BY-NC 4.0; https://creativecommons.org/licenses/by-nc/4.0/), once access to the datasets in the Genomics England Research Network and Genes and Health Research Networks is confirmed. As many codes work on individual-level data, they need to be imported into the relevant trusted research environments to perform the analyses, with the results exported from the TREs subject to the individual TRE restrictions.

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

## Acknowledgements

This work was supported by Barts Charity (MGU0504) to C.C. and L.J.J. and Barts NIHR BRC (BTXH1A1R) to C.C. and L.J.J. BCN Biobank is funded by the Breast Cancer Now charity (TB2022BAR) to C.C. and L.J.J. This work forms part of the research portfolio of the National Institute for Health and Care Research Barts Biomedical Research Centre (NIHR203330); a delivery partnership of Barts Health NHS Trust, Queen Mary University of London, St George's University Hospitals NHS Foundation Trust and St George's University of London. This research was made possible through access to data in the National Genomic Research Library, which is managed by Genomics England Limited (a wholly owned company of the Department of Health and Social Care). The National Genomic Research Library holds data provided by patients and collected by the NHS as part of their care and data collected as part of their participation in research. The National Genomic Research Library is funded by the National Institute for Health Research and NHS England. The Wellcome Trust, Cancer Research UK and the Medical Research Council have also funded research infrastructure. The results published here are in part based upon data generated by the TCGA Research Network[19]. The authors wish to acknowledge the roles of the Breast Cancer Now Biobank in collecting and making available the samples and/or data, and the patients who have generously donated their tissues and shared their data to be used in the generation of this publication. We are especially grateful to members of the BCN Biobank-BCI (Catherine McMaster-Christie, Rachel Nelan, Jennifer McGuinness, Jenny Gomm and Iain Goulding) for their help in setting up the framework for data collection. We thank Barts Health NHS Trust for their help with the collection of secondary and tertiary care data for BCN Biobank. We thank members of the Discovery East London Programme Board, Discovery Data Service/Endeavour Health Charitable Trust and Voror Health Technologies Ltd for their support in facilitating collection of BCN Biobank primary care patient records. Genes & Health has recently been core-funded by Wellcome (WT102627, WT210561), the Medical Research Council (UK) (M009017, MR/X009777/1, MR/X009920/1), Higher Education Funding Council for England Catalyst, Barts Charity (845/1796), Health Data Research UK (for London substantive site), with research delivery support from the NHS National Institute for Health Research Clinical Research Network (North Thames). Most of all, we thank all of the individuals participating in the Genomics England 100,000 Genomes Project, The Cancer Genome Atlas, Genes & Health and the Breast Cancer Now Biobank.

## Author contributions

Methodology: G.J.T., E.G., C.C.; Formal Analysis: G.J.T., E.G., A.Z.M.D.U., L.G.E.J.; Data access and Curation: G.J.T., E.G., A.Z.M.D.U., M.A., R.B.-M., L.J.J., C.C.; Visualisation: G.J.T. and E.G.; Writing—original draft: G.J.T., E.G.; Writing—revisions: G.J.T., E.G., L.G.E.J.; Writing—review and editing: all authors; Funding acquisition: L.J.J. and C.C.; Supervision and Conceptualisation: C.C. All authors read and approved the final manuscript.

## Competing interests

The authors declare no competing interests.
