## [Peer Review File · Nature Communications]

The clinical and molecular landscape of breast cancer in women of African and South Asian ancestry

Corresponding Author: Professor Claude Chelala

Version 0:

Reviewer comments:

Reviewer #1

(Remarks to the Author)

Enclosed is a review of "The clinical and molecular landscape of breast cancer in women of African and South Asian ancestry" by Claude Chelala's group. In this manuscript, the authors employed a multi-faceted approach to cataloging how the clinical and molecular architecture of breast cancer in women of African and South Asian ancestry with primary diagnoses compares to that of European-ancestry women. The authors used data from four databases collected across the UK and US and identified multiple clinical phenotypes, somatic variants, and germline variants differentially distributed amongst these two genetic ancestry groups compared to Europeans. Overall, the manuscript is well-written and helps address the underrepresentation of these ancestral groups in the breast cancer literature. It would benefit from greater detail and justification of the statistical methods used, as well as direct analyses comparing the African and South Asian groups themselves.

Major Comments

1. Despite the results of the general concordance of SRE and gAncestry in the Genomics England and TCGA cohorts (Figure 3), there may be concern over the use of SRE instead of gAncestry in analyses of the BCN Biobank Barts (N=1895). As indicated in guidelines developed in response to the 2023 NASEM report on the use of population descriptors (ref 1), self-identified race/ethnicity should not be used as a proxy for genetic ancestry. This is particularly important when analyses of the remaining cohorts does use genetically inferred ancestry as the primary independent variable. Please also clearly distinguish between race, ethnicity, and self-reported measures of these (e.g., Supplemental Figure 2a).
2. Is there admixture present in SAS British participants from the G&H cohort, and why were admixed cohorts excluded when the sample sizes are relative comparable in Genomics England (N=128) and TCGA (N=125)? There exist several statistical methods designed to handle analyses of admixed subjects. It may be interesting to also include analyses incorporating local ancestry instead of global ancestry assignment. For example, it would be informative to see how the identified mutations' frequencies differ by level of admixture, specifically at the genomic locus of these identified mutations.
3. In the logistic/linear regression models, what covariates were adjusted for? More detail and justification should be provided for all statistical analyses. It may be advantageous to model AFR, EUR, and SAS jointly to allow for direct AFR vs. SAS comparisons across all phenotypes considered.
4. What about germline genomic variant data for TCGA? What are the sample sizes? Regardless, the authors need to mention what the sample sizes are in cohorts besides Genomics England.
5. I have a few questions about how ancestry-corrected tumor mutational burden is computed. Does this remove effects of ancestry before the comparison of AFR and SAS vs. EUR? If so, the authors should address how this quantity can be modeled properly.
6. For germline variant profiles, why did authors not consider all genes? Why were analyses limited to just those genes with previously exhibited ancestry-associated differences?
7. Figure 7 can benefit from some more details regarding on the germline/somatic effects change the genes and molecular pathways shown in the middle panel. Simply showing red and blue lines for what I assume are positive and negative effects are hard to interpret.
8. I wonder if stratifications on ancestry in the survival analyses may bring in some collider bias. I suggest some exploration of this through either sensitivity analyses or related methods, like SlopeHunter (see <https://pubmed.ncbi.nlm.nih.gov/36821633/>).
9. This may be a naïve question, but why is OLS with the log mutational burden the norm here, instead of a count regression

like a Poisson/negative binomial regression?

Minor Comments

10. Please add line numbers for ease of review.
11. Last paragraph of page 2 -- 'highlight' should be 'highlights'.
12. Bottom of page 3 – define NHS patients.
13. Inconsistent use of BC abbreviation (e.g., introduction).
14. gAncestry is not defined prior to use in legend of Figure 1. Same for SAS.
15. Top of page 7 -- $3334+2479+1076+363=7252$, not 7136. Should indicate this is the total of those subjected included, not recruited.
16. Supplemental Figure 2 – 'American' and 'Other' are the same color. This might be confusing to readers.
17. Abbreviations (e.g., IMD) not defined before mentioned in Supplemental Table 1.
18. Use of "covariate" on page 10 is confusing and on subsequent pages, e.g. middle of 11.
19. All p-values from fitted linear/logistic regression models should be provided in addition to those from univariate descriptive tests.
20. Page 11 – use of p.adj versus p in previous paragraphs.
21. Page 11 – Authors determined that a full model including IMD showed a significant improvement in fit for age at diagnosis. It would be informative to present test statistics and p-values for the gAncestry coefficients in this model instead (supplement).
22. Supplemental Fig 4, page 12 – What are the sample sizes for these age-stratified analyses?
23. Font of Figure 4b-c is not readable.
24. Figure 4a – Are these the results for Genomics England only? Need to confirm in caption.
25. Figure 5 – What does white space (with percentage label) separating bars indicate? Significance? Also, the caption says these were from linear regression.
26. Why the cutoff of OR p-value of 0.1 for "differentially present" on page 28? Reference for this? Similarly, why 2% cut off for threshold method? Need some sort of justification. Why not look at variants present in AFR + SAS that are absent in EUR?
27. Figure 6 – Define blue and red in the caption.
28. Page 19 – First paragraph, mention somatic. Also, I hesitate to call this "validation" in the section header.
29. Genes should be italicized.
30. Mutational signatures/HR deficiency analyses done only in Genomics England? (double check)
31. Any relatedness QC done on these datasets?
32. For demographic and clinical features analysis (page 27), need to explicitly detail linear and logistic regression models. Why not try multinomial logistic regression model for unordered factors of multiple levels?
33. Page 28 – VAF $\geq 3\%$ or $< 3\%$?
34. Page 29 – Where are the lollipop lots mentioned? (double check)

(1) Feero WG, Steiner RD, Slavotinek A, et al. Guidance on Use of Race, Ethnicity, and Geographic Origin as Proxies for Genetic Ancestry Groups in Biomedical Publications. *JAMA*. 2024;331(15):1276–1278. doi:10.1001/jama.2024.3737

Reviewer #2

(Remarks to the Author)

The manuscript: "The clinical and molecular landscape of breast cancer in women of African and South Asian ancestry", by Thorn and collaborators describes the molecular characterization of breast cancer from African and South Asian patients, compared to European patients.

The manuscript confirms well known observations regarding the lack of ethnic representation in cancer genomics databases worldwide and presents compelling evidence on why it is necessary to increase diversity representation in genomics databases to deliver the benefits of precision medicine to a much higher number of patients around the world.

The manuscript is also a very good example of the use and application of existing public databases to answer both clinical research questions and pinpoint important health disparities. This is particularly useful when budget for genomic characterization is scarce.

However, there are some questions that, in my opinion, must be addressed by the authors before publication.

- 1) My main concern is that all comparisons seem to be made using mutation calling data from each one of the databases, however, there is no discussion about potential differences in the genes and variants called by each of the mutation calling algorithms and how this might affect direct comparisons. The authors must clarify this issue early in the manuscript, since this point is the cornerstone of the whole paper.
- 2) Even though the authors identified several differentially mutated genes in the populations they analyzed, the work would be greatly improved if validation of these mutation frequencies are validated in an independent cohort of South Asian patients. This will indeed represent time and money, but focusing on validating a limited number of mutations through direct genotyping would really validate the observations made through database analysis. The same for germline mutations.
- 3) Figure 3 might be considered as supplementary.

4) A number of genes with differentially mutated frequencies were identified, however, the discussion about how they might play a role in the observed clinical differences between ethnic groups is not really there. Observations regarding an earlier disease presentation with more aggressive tumors in the African population confirms previous reports. This was also observed in South Asian patients, however, SA patients show a lower degree of tumor mutational burden, would any of the differentially mutated genes play a role in this observation?

5) Regarding the clinical actionability of the mutated genes, is there any follow up information that would indicate if some of these patients received any targeted therapy and how it modified clinical response?

Reviewer #3

(Remarks to the Author)

The goal of this study is to describe the clinical and molecular landscape of breast cancer among women with African and South Asian ancestries. The study is motivated by the fact that existing studies have been predominantly performed in European populations and health disparities in breast cancer risk and progression could be exacerbated if precision medicine fails to include a representative panel of therapeutic and prognostic targets. The study recapitulates much of what is known regarding HR-deficiency in African compared to European populations and identifies differential mutation of genes included in current clinical assays. While the authors do identify additional genes and variants differentially mutated that are not included in current clinical assays, these findings suffer from a lack of clarity in presentation and focus. While the results from this analysis do have merit, the authors' argument for "germline screening protocols modified based on ethnicity" is not well supported by the current study.

1. Descriptive Statistics of Clinical and Molecular Features

- a. The authors describe a bimodal age distribution for breast cancer incidence in the EUR population and unimodal distributions in the AFR and SAS populations. However, it appears that all three populations may have overlapping Gaussian distributions and may all be multi-modal.
- b. Detailed attention is given to measures of deprivation. However, this measure does not feature heavily in the main analyses of the paper other than its inclusion as a covariate. It could be incorporated into the analyses for somatic mutations and somatic mutational signatures. Minor: The discussion on allostatic load and breast cancer features needs to be supported by references from the literature.

2. Somatic Variants in the Genomics England Cohort

- a. The authors identified differences in mutational frequencies for genes in the AFR and SAS Genomics England Cohorts; 6 of these genes were also differentially mutated between AFR and EUR populations in TCGA. However, it is unclear whether the comparison to the TCGA cohort is based on results from a logistic or threshold model. Further the presentation of both approaches for the GEC cohort makes it somewhat difficult to follow the text. The authors appear to emphasize the logistic approach and thus may consider presenting these results only to improve clarity.
- b. Differential mutation frequencies were identified among 7 variants in the AFR cohort and 4 variants in the SAS cohort from GEC. Figure 5 could benefit from labeling to indicate areas of the graph referring to genes vs. variants.

3. Germline Variant Profiles of Cancer Susceptibility Genes + Validation

- a. The authors state that 6 genes that were differentially mutated between the EUR and AFR cohorts were also differentially mutated in TCGA. The six genes mentioned are not featured within a primary table or figure for the manuscript but are included in supplementary text. This makes it difficult to follow the author's interpretation of the findings. Further, the authors toggle between discussions for somatic and germline mutations and should clearly state which they are referring to in the relevant text (see page 19, first paragraph).

Reviewer #4

(Remarks to the Author)

Version 1:

Reviewer comments:

Reviewer #2

(Remarks to the Author)

The answers provided by the authors are clear and satisfactory. In my opinion, the paper might be accepted.

Reviewer #4

(Remarks to the Author)

The authors have largely addressed all of my previous comments regarding this manuscript. I appreciate the detailed

explanation of genetic ancestry within the cohorts, particularly as it pertains to admixture, as well as the interesting comparison of HRdetect between WGS and WXS in TCGA. I additionally think that the updated mutational burden analysis incorporating a negative binomial model and expanded gene set for the germline variant profiles are improvements. A minor recommendation would be, pertaining to my comment #26, to add the reference justifying this cut-off value to the manuscript body itself.

Response to Reviewers

Reviewer #1

Enclosed is a review of “The clinical and molecular landscape of breast cancer in women of African and South Asian ancestry” by Claude Chelala’s group. In this manuscript, the authors employed a multi-faceted approach to cataloging how the clinical and molecular architecture of breast cancer in women of African and South Asian ancestry with primary diagnoses compares to that of European-ancestry women. The authors used data from four databases collected across the UK and US and identified multiple clinical phenotypes, somatic variants, and germline variants differentially distributed amongst these two genetic ancestry groups compared to Europeans. Overall, the manuscript is well-written and helps address the underrepresentation of these ancestral groups in the breast cancer literature. It would benefit from greater detail and justification of the statistical methods used, as well as direct analyses comparing the African and South Asian groups themselves.

Major Comments

1. Despite the results of the general concordance of SRE and gAncestry in the Genomics England and TCGA cohorts (Figure 3), there may be concern over the use of SRE instead of gAncestry in analyses of the BCN Biobank Barts (N=1895).

As indicated in guidelines developed in response to the 2023 NASEM report on the use of population descriptors (ref 1), self-identified race/ethnicity should not be used as a proxy for genetic ancestry. This is particularly important when analyses of the remaining cohorts does use genetically inferred ancestry as the primary independent variable.

Please also clearly distinguish between race, ethnicity, and self-reported measures of these (e.g., Supplemental Figure 2a).

We thank the reviewer for the reference and agree that race, ethnicity, and geographic origin should not be used as proxies for genetic ancestry (gAncestry) groups due to their discrete and continuous natures, respectively (1). We apologise for the wording of the manuscript in which this may have been unclear and have amended the text accordingly (Results pp4-10, lines 71-222; Discussion pp25-26, lines 555-580).

Genetic ancestry became available for the Genes and Health (G&H) breast cancer cohort after the initial submission. This allowed us to update our analyses and results (Revised Figures 2 and 3; Supplementary Figure 1c), and eliminate self-reported measures as an analytical parameter for this cohort. Genetic data is not available for patients in the BCN Biobank analytical cohort (n=2,126), except for 231 patients dually consented by both the Biobank and Genomics England. For these patients, it is not possible to derive measures of gAncestry and, as such, self-identified ethnicity was used to stratify patients. We can confirm that ethnicity is not used as an independent variable from which our findings are defined and that these are derived from genetically inferred and biological/clinical covariates.

(1) Feero WG, Steiner RD, Slavotinek A, et al. Guidance on Use of Race, Ethnicity, and Geographic Origin as Proxies for Genetic Ancestry Groups in Biomedical Publications. *JAMA*. 2024;331(15):1276–1278. doi:10.1001/jama.2024.3737

2a. Is there admixture present in SAS British participants from the G&H cohort, and why were admixed cohorts excluded when the sample sizes are relatively comparable in Genomics England (N=128) and TCGA (N=125)? There exist several statistical methods designed to handle analyses of admixed subjects.

2b. It may be interesting to also include analyses incorporating local ancestry instead of global ancestry assignment. For example, it would be informative to see how the identified mutations' frequencies differ by level of admixture, specifically at the genomic locus of these identified mutations.

2a. The relative admixture proportions are not currently available from the G&H research environment. A total of 340/364 breast cancer patients have gAncestry determinations now available to us from G&H research environment. This information also includes the classification of the superpopulation and sub-South Asian population for each individual. These sub-South Asian populations represent aggregates of those defined by the 1000 Genomes (used by Genomics England (Reviewer Figure 1). For example, the Pakistani subpopulation clusters with the Punjabi (PJL), Gujarati (GIH) and some Tamil (STU) participants in 1000 Genome and the Bangladeshi subpopulation gAncestry in G&H clusters with the Bengali (BEB), Telugu (ITU) and some Tamil (STU) participants within the 1000 Genomes.

Reviewer Figure 1. (From G&H Documentation) **a.** Scatterplot of first 2 PCs of ancestry determinants from G&H (in blue) against the five superpopulations of the 1000 Genomes; **b.** UMAP derived from first 7 PCs of the G&H (elgh, in blue) against the five SAS subpopulations of the 1000 Genomes. Here, the SAS population are categorised into two clear groups: the Pakistani subpopulation comprising the Punjabi (PJL), Gujarati (GIH) and some Tamil (STU) participants and Bangladeshi subpopulation comprising the Bengali (BEB), Telugu (ITU) and some Tamil (STU) participants within the 1000 Genomes Project.

The Admix cohorts in Genomics England and TCGA were excluded because investigations showed that this group (Reviewer Figure 2, in orange) did not contain a set of participants with direct admixture between two of our major ancestry groups:

Reviewer Figure 2. Scatterplots of the relative EUR admixtures within the Admix population.

2b. While, it is beyond the scope of this initial project, we agree that examining the local diversity in genetic ancestry would be a very interesting expansion of this research such as in Martini et al (2).

Looking at the **superpopulations** of our three ancestry cohorts within Genomics England we find that more than 70% of these have an assignment of 1.0 to a superpopulation. This restricted admixture gradient limits the ability to examine the genetic diversity within superpopulations. Furthermore, the pairwise ancestry fractions also did not directly show admixture between our major cohorts (Reviewer Figure 2).

Looking at the **Admix population**, we originally presented the superpopulation contributions per patient (Supplementary Figure 2). The Genomics England dataset also has subpopulation contributions per patient available. However, this subpopulation assignment used a different set of SNPs, filtered using a different minor allele cut off (0.01 for the subpopulations, compared to 0.05 for the superpopulations: https://re-docs.genomicsengland.co.uk/ancestry_inference/), thus the subpopulation contributions do not sum to the superpopulation contributions per admixed participant. Although the same general trends can be seen (e.g., the AFR admix group appears to have the most contributions from the AFR subpopulations), some participants would be re-assigned to the main superpopulation ancestry groups if the subpopulations were used (Reviewer Figure 3).

As such, we do not have enough participants within the main ancestry groups and the Admix group in which to generate the admixture gradient for testing variant frequencies, as suggested by reviewer 1. We agree that it would be a fruitful further avenue of research within the UK population. Similarly, (subpopulation) ancestry proportions are not available for G&H at this time, but, with the ability to generate case:control studies within this cohort, this would be an exciting research direction in this underrepresented population.

Reviewer Figure 3. Admixture contributions within the Admix population. **a.** at the superpopulation level, **b.** with the assigned subpopulations.

(2) Martini R, Delpé P, Chu TR, Arora K, Lord B, Verma A, Bedi D, Karanam B, Elhussin I, Chen Y, Gebregzabher E, Oppong JK, Adjei EK, Jibril Suleiman A, Awuah B, Muleta MB, Abebe E, Kyei I, Aitpillah FS, Adinku MO, Ankomah K, Osei-Bonsu EB, Chitale DA, Bensenhaver JM, Nathanson DS, Jackson L, Petersen LF, Proctor E, Stonaker B, Gyan KK, Gibbs LD, Monojlovic Z, Kittles RA, White J, Yates CC, Manne U, Gardner K, Mongan N, Cheng E, Ginter P, Hoda S, Elemento O, Robine N, Sboner A, Carpten JD, Newman L, Davis MB. African Ancestry-Associated Gene Expression Profiles in Triple-Negative Breast Cancer Underlie Altered Tumor Biology and Clinical Outcome in Women of African Descent. *Cancer Discov.* 2022 Nov 2;12(11):2530-2551. doi: 10.1158/2159-8290.CD-22-0138. PMID: 36121736; PMCID: PMC9627137.

3. In the logistic/linear regression models, what covariates were adjusted for?

More detail and justification should be provided for all statistical analyses. It may be advantageous to model AFR, EUR, and SAS jointly to allow for direct AFR vs. SAS comparisons across all phenotypes considered.

For Revised Figure 3 in the revised manuscript, we did not adjust for any covariate, but we tested nested submodels, which included the Index of Multiple Deprivation quintile (IMD) *via* likelihood ratio tests, and found that age at diagnosis was the only factor where IMD was a significant confounder.

All three ancestry groups were modelled simultaneously for the clinical covariates. In response to this comment, we have also run the AFR v SAS ancestry direct comparison and found that the computed effect sizes were equal to the differences between the EUR v SAS and EUR v AFR comparisons (Results, section: gAncestry-Associated Somatic Variants within the Genomics England Cohort, pp10-11, lines 224-245).

We agree with the reviewer that the AFR/SAS direct comparison warrants further investigation, but we feel that we require larger datasets in both ancestries to obtain sufficient power to determine significant differences in somatic and germline variation between the two groups. We are looking into larger validation datasets (including The India Cancer Genome Atlas and Marengo Asian Hospital India, with whom we are now exploring future collaborations), but a full analysis is beyond the scope of this manuscript.

4. What about germline genomic variant data for TCGA? What are the sample sizes? Regardless, the authors need to mention what the sample sizes are in cohorts besides Genomics England.

6. For germline variant profiles, why did authors not consider all genes? Why were analyses limited to just those genes with previously exhibited ancestry-associated differences?

Due to the similar nature of comments 4 and 6, we have responded to these simultaneously. We thank the reviewer for these comments, and we believe that addressing these has strengthened the manuscript considerably.

In the original manuscript, we considered genes that are currently tested in the clinic, regardless of ancestry, as well as those reported to exhibit ancestry-associated differences in the literature. We agree that this limited the breadth of our findings and could be biased towards those genes most reported in the literature. As such, we expanded the analysis to include 180 genes: those tested in the clinic (as reported in the NHS Genomic Medicine Service Panels v8); and those conferring susceptibility to cancer (as reported in the Genomics England PanelApp gene panel, <https://panelapp.genomicsengland.co.uk/>).

We modified the germline analysis workflow to allow for its application to TCGA and G&H cohorts in a reproducible and transparent manner. The germline section (Main Figure 5, pp11-12, lines 247-279) now provides an overview of relative contributions of pathogenicity classifications within these genes, before focusing on those variants with a high level of evidence to support their association to cancer.

Revised Figure 5 shows the comparison between Genomics England and TCGA for Germline, the variant prioritisation workflow (Revised Figure 9), with genes differentially mutated between EUR and non-EUR cohorts in Genomics England and in TCGA, and in the G&H case:control study in Supplementary Table 8.

5. I have a few questions about how ancestry-corrected tumor mutational burden is computed. Does this remove effects of ancestry before the comparison of AFR and SAS vs. EUR? If so, the authors should address how this quantity can be modeled properly.

We adjust for ancestry prior to the comparison between ancestries to remove the potential effects of ancestry which are considered 'normal' within these groups. Nassar et al (3), demonstrated the benefit of administering immune checkpoint inhibitors to patients with high TMB; a finding that was only apparent when this was corrected for genetic ancestry. This highlights the translational importance of ancestry-based determination of TMB.

We have updated the text and added a flow chart to better describe the filtering process applied for the TMB calculation (Main Figure 8 and Methods pp21-22, lines 484-500).

(3) Nassar AH et al., Ancestry-driven recalibration of tumor mutational burden and disparate clinical outcomes in response to immune checkpoint inhibitors. *Cancer Cell* 2022; 40(10):1161-1172.

7. Figure 7 can benefit from some more details regarding on the germline/somatic effects change the genes and molecular pathways shown in the middle panel. Simply showing red and blue lines for what I assume are positive and negative effects are hard to interpret.

The text and figure have been updated to enhance clarity.

8. I wonder if stratifications on ancestry in the survival analyses may bring in some collider bias. I suggest some exploration of this through either sensitivity analyses or related methods, like SlopeHunter (see <https://pubmed.ncbi.nlm.nih.gov/36821633/>).

With such a small number of genetic features, applying a method such as SlopeHunter (primarily used for GWAS associations) to reduce the potential effects of collider bias is not feasible within the scope of this manuscript. However, we have added a section in the discussion (p30, lines 666-677) to address potential colliders into the discussion.

9. This may be a naïve question, but why is OLS with the log mutational burden the norm here, instead of a count regression like a Poisson/negative binomial regression?

We originally attempted a scaled Poisson model but the dispersion in the distribution of variant counts was too large for the dispersion model within a Poisson regression. A log-scaled OLS model for the TMB removed this restriction by separately modelling the mean and the variance of the distributions. However, this is still not a perfect model: and so, we thank the reviewer for suggesting the negative binomial model which is better suited to count data. We have updated the analysis to use a negative binomial model applied to the raw exome mutation count data, which allowed a better point estimate of the effects.

An example between the OLS model on the log(TMB) and the scaled negative binomial model on the scaled TMB is shown below.

Original: linear least squares on log (TMB) differences between AFR and EUR ancestry groups in Genomics England

New: generalised linear model using scaled negative binomial regression on same dataset

The forest plots have been updated with the new model.

Minor Comments

10. Please add line numbers for ease of review.

Apologies. Line numbers have been added to the manuscript.

11. Last paragraph of page 2 -- 'highlight' should be 'highlights'.

We thank the reviewer for identifying this typo; the text has been updated.

12. Bottom of page 3 – define NHS patients.

We have amended the text to incorporate this comment.

13. Inconsistent use of BC abbreviation (e.g., introduction).

We thank the reviewer for identifying this inconsistency; this has been addressed throughout the text.

14. gAncestry is not defined prior to use in legend of Figure 1. Same for SAS.

The legend of Revised Figure 1 has been updated to ensure that gAncestry and SAS are defined prior to their use.

15. Top of page 7 -- 3334+2479+1076+363=7252, not 7136. Should indicate this is the total of those subjected included, not recruited.

Thank you for identifying this inconsistency, we have updated values to 7,253 (3334 (Genomics England) + 2479 (BCN) + 1076 (TCGA) +364 (G&H)). Figure 2 has also been updated to present this value more clearly.

16. Supplemental Figure 2 – 'American' and 'Other' are the same color. This might be confusing to readers.

We agree with the reviewer and have re-generated the figures ensuring contrast between colours.

17. Abbreviations (e.g., IMD) not defined before mentioned in Supplemental Table 1.

The subheading in Supplemental Table 1 has been updated.

18. Use of “covariate” on page 10 is confusing and on subsequent pages, e.g. middle of 11.

We have updated the text to decrease any confusion.

19. All p-values from fitted linear/logistic regression models should be provided in addition to those from univariate descriptive tests.

P-values are available in the relevant Supplementary Tables (Supplementary Tables 1b-c , 2b, 5b, 7 and 8)

20. Page 11 – use of p.adj versus p in previous paragraphs.

We have reverted to p-val in this paragraph.

21. Page 11 – Authors determined that a full model including IMD showed a significant improvement in fit for age at diagnosis. It would be informative to present test statistics and p-values for the gAncestry coefficients in this model instead (supplement).

The test statistics relating to IMD have been added to the main text and supplementary information in Supplementary Table 3.

22. Supplemental Fig 4, page 12 – What are the sample sizes for these age-stratified analyses?

The sample sizes have been clarified both in the text and the Supplementary Figure (now Supplementary Figure 5).

23. Font of Figure 4b-c is not readable.

Apologies for this oversight. The font has been amended to ensure readability.

24. Figure 4a – Are these the results for Genomics England only? Need to confirm in caption.

The text has been updated in the caption.

25. Figure 5 – What does white space (with percentage label) separating bars indicate? Significance? Also, the caption says these were from linear regression.

We have redrawn the figure (now Revised Figure 4) to make this clearer and updated the caption.

26. Why the cutoff of OR p-value of 0.1 for “differentially present” on page 28? Reference for this?

look at variants present in AFR + SAS that are absent in EUR?

We reused the adjusted p-value cut-off used in the methods of Carrot-Zhang et al. paper, where they used this as an initial filter prior to validation and model-fitting.

As the focus of the manuscript was on the pairwise comparisons EUR v AFR and EUR v SAS, we did not look at combining the non-EUR populations together, either for the logistic model or for the 2% threshold model.

27. Figure 6 – Define blue and red in the caption.

This figure has been changed in response to comment 4, in which we also display information from TCGA.

28. Page 19 – First paragraph, mention somatic. Also, I hesitate to call this “validation” in the section header.

The section header has been amended.

29. Genes should be italicized.

Gene names are now in italics in the revised manuscript and associated figures and tables.

30. Mutational signatures/HR deficiency analyses done only in Genomics England? (double check)

We only originally performed mutational signature and HR deficiency analysis using the data within Genomics England because this was a complete set of WGS data, including all short variants (SNPs and indels), longer structural variants (SVs) and copy number changes (CNVs).

The data available for the TCGA BRCA cohort consists of mostly WES data with the remainder (102 participants) with a similarly complete set of WGS data. We confirmed this with the Genomics Data Commons. Following the method in this preprint: <https://www.medrxiv.org/content/10.1101/2024.07.14.24310383v1.full>, one approach with WES data is to compute the scores for the SNP, indel and copy number change signatures as per the HRDetect algorithm, but set both the SV3 and SV5 signatures to zero, and compute the HR deficiency probability this way. However, the authors of the HRDetect algorithm (<https://github.com/Nik-Zainal-Group/signature.tools.lib>) caution against this as the parameters of the logistic classifier are not specific to WES data, thus meaning that the HR deficiency score may be inaccurate.

We ran our own test as to HRDetect feasibility on WES data by computing the score for the 88 female participants with WGS data on the original WGS data and a filtered dataset simulating WES data for the same participants (Reviewer Figure 4). If we take the WGS data as a ground truth, we find that 6 of our samples were misclassified as being HR proficient (HRDetect score < 0.7) when considering the WES data, but the other 82 were classified correctly. The scores themselves show considerable scatter, with many samples close to being misclassified, which indicate that it is infeasible to compute the score (using the WES modifications to the algorithm) using the whole TCGA BRCA cohort because of the risk of misclassification.

Reviewer Figure 4: HRDetect score for WGS participants on unfiltered WGS data and filtered (synthetic) WES data

If we were to restrict our attention to the 88 WGS female participants, the gAncestry split is for 73 EUR, 4 AFR and 11 East Asian, which means that we would not have enough statistical power to detect any differences between the two main groups EUR v AFR for direct comparison with our work on the Genomics England cohort.

31. Any relatedness QC done on these datasets?

TCGA and BCN Biobank patients were/are recruited under strict selection criteria from multiple sites through their workflows and are known to be unrelated.

Genomics England performed relatedness QC using PLINK (<https://www.cog-genomics.org/plink/>) and KING (<https://www.kingrelatedness.com>) throughout all their participants because of the rare disease component of the 100K Genomes Programme. For this, participants with rare diseases were genotyped, along with their parents and other relatives who may have been affected. Within the Genomics England breast cancer cohort, less than 3% of the individuals have kinship coefficients > 0.0442 with other participants within the same breast cancer cohort, which is indicative of closer than third-degree relationships.

Genes and Health ran the same method for their whole 51k cohort, which when we performed the filtering down to the 17922 participants in our case and control cohorts, 14% (2511) of these individuals had closer than second-degree relationships with other members of the combined G&H groups. This number is higher due to the recruitment methods and criteria used in the G&H programme.

We did not run any filtering for relatedness on the Genomics England and G&H cohorts as we did not run a full germline (or somatic) GWAS on these cohorts.

32. For demographic and clinical features analysis (page 27), need to explicitly detail linear and logistic regression models. Why not try multinomial logistic regression model for unordered factors of multiple levels?

We have detailed the linear and logistic models for the demographic and clinical factors in the Extended Methods (p7). We feel that there is no appropriate method for fitting a multinomial logistic regression model, as our predictor variable (ancestry) is not a continuous variable and only takes two values, 0 and 1. In addition, the factors with multiple levels (such as the IMD quintile) are ordered.

33. Page 28 – VAF $\geq 3\%$ or $< 3\%$?

The wording of this paragraph has been amended for clarity (Methods p21, lines 476-482) and we have added a flow chart/decision tree (Revised Figure 8) to clarify the point further.

34. Page 29 – Where are the lollipop lots mentioned? (double check)

We apologise, but the lollipop plots were in a previous version of the manuscript, and the reference should have been removed before submission. This has been changed this to reflect the updated figures.

Reviewer #2 (remarks to the author)

The manuscript: “The clinical and molecular landscape of breast cancer in women of African and South Asian ancestry”, by Thorn and collaborators describes the molecular characterization of breast cancer from African and South Asian patients, compared to European patients.

The manuscript confirms well known observations regarding the lack of ethnic representation in cancer genomics databases worldwide and presents compelling evidence on why it is necessary to increase diversity representation in genomics databases to deliver the benefits of precision medicine to a much higher number of patients around the world. The manuscript is also a **very good example of the use and application of existing public databases** to answer both clinical research questions and pinpoint important health disparities. This is particularly useful when budget for genomic characterization is scarce. However, there are some questions that, in my opinion, must be addressed by the authors before publication.

1) My main concern is that all comparisons seem to be made using mutation calling data from each one of the databases, however, there is no discussion about potential differences in the genes and variants called by each of the mutation calling algorithms and how this might affect direct comparisons. The authors must clarify this issue early in the manuscript, since this point is the cornerstone of the whole paper.

We thank the reviewer for this important observation. Much of the data that we have used has already been extensively processed through variant calling pipelines and is available from distinct trusted research environments via specific data access agreements. We have amended the text to add a brief discussion and comparison of the different calling methods used. (Discussion, p25, lines 546-554)

2) Even though the authors identified several differentially mutated genes in the populations they analyzed, the work would be greatly improved if validation of these mutation frequencies are validated in an independent cohort of South Asian patients. This will indeed represent time and money, but focusing on validating a limited number of mutations through direct genotyping would really validate the observations made through database analysis. The same for germline mutations.

We agree with the reviewer that the manuscript would be improved using an independent cohort to compare the results from the South Asian patients. To this end, we performed a germline case:control study on the G&H population, where the cases were those confirmed to have South Asian gAncestry and a diagnosis of breast cancer, and the controls were a gender and gAncestry-matched population from the cancer-free individuals of the G&H cohort (Supplementary Table 8).

We are also envisaging and exploring future collaborations with ICGA (The India Cancer Genome Atlas) and Marengo Asia Hospitals (5 Hospitals in India) to increase the size of our SAS cohorts.

3) Figure 3 might be considered as supplementary.

Figure 3 has been moved into the Supplementary figures, as Supplementary Figure 1.

4) A number of genes with differentially mutated frequencies were identified, however, the discussion about how they might play a role in the observed clinical differences between ethnic groups is not really there.

Observations regarding an earlier disease presentation with more aggressive tumors in the African population confirms previous reports. This was also observed in South Asian patients, however, SA patients show a lower degree of tumor mutational burden, **would any of the differentially mutated genes play a role in this observation?**

There is a higher proportion of ER-/HER2- cancers in both the AFR and SAS Genomics England cohorts, and an increased prevalence of germline *BRCA1/2* mutations (and mutations in other DNA damage repair genes), meaning that cancers in these two populations would be more aggressive, even with lower somatic TMB, such as in the SAS cohort.

We found no association of the differentially mutated somatic genes in the SAS cohort with disease aggressiveness.

5) Regarding the clinical actionability of the mutated genes, is there any follow up information that would indicate if some of these patients received any targeted therapy and how it modified clinical response?

While our project was not designed as a treatment-effect study we agree that integrating pharmacogenomic evaluations onto longitudinal clinical data is a very interesting and is the focus of the next study in preparation.

Research has reported on the clinical efficacy of including PARP inhibitors (PARPi), either as a single agent or in combination with existing treatment regimens, in ameliorating outcome in breast cancer patients with germline *BRCA1/2* and *PALB2* mutations.

We expanded upon our pharmacogenomic section based on two main criteria: first, to identify therapeutic opportunities in patients that could have potentially benefitted from the inclusion of PARPi in their treatment; second, to identify potential windows of resistance to specific treatments in HR deficient patients.

To ensure robustness of our findings we limited our new analysis to patients with long-term clinical follow-up that were dually consented by both Genomics England and the Breast Cancer Now Biobank. For these patients, we were able to access both genomic data and up-to-date longitudinal health care data directly from Barts Health NHS Trust hospital electronic health records (EHRs) (both structured and unstructured free-text data) thus offering extensive granularity of longitudinal information. We then focussed on those patients from EUR, AFR and SAS gAncestry with longitudinal healthcare data to best represent the analytical cohort used in the manuscript (n=160).

We identified that an eighth of patients (12.50%) that were not administered PARPi as a part of their clinical journey had pathogenic germline mutations in *BRCA1/2* (Revised Figure 7b, Results p15, lines 343-352). Furthermore, when including then presence of germline *PALB2*, a gene mutated in significantly higher proportions in the AFR population relative to EUR, which also confers susceptibility to PARPi, the proportion of patients that may have benefitted from inclusion of PARPi as a part of treatment increased to 31.25% (Revised Figure 7c, Results p15 lines 343-352).

We identified 17 (10.63%) patients as HR deficient and, as such, may show propensity to resistance to taxane therapies. When looking at the clinical journey of these patients, 70.56% were administered therapeutic regimens inclusive of a taxoid agent, of which 47.1% had a secondary event indicative of progression (e.g., recurrence, metastasis and death) (Revised Figure 7a, Results p15, 337-342).

While the mutations and dysfunctions discussed above were identified as differentially mutated in non-EUR cohorts, the clinical implications of these results are not specific to a single ethnicity and, instead, are of benefit to all patients.

Reviewer #3 (remarks to the author)

The goal of this study is to describe the clinical and molecular landscape of breast cancer among women with African and South Asian ancestries. The study is motivated by the fact that existing studies have been predominantly performed in European populations and health disparities in breast cancer risk and progression could be exacerbated if precision medicine fails to include a representative panel of therapeutic and prognostic targets. The study recapitulates much of what is known regarding HR-deficiency in African compared to European populations and identifies differential mutation of genes included in current clinical assays. While the authors do identify additional genes and variants differentially mutated that are not included in current clinical assays, these findings suffer from a lack of clarity in presentation and focus. While the **results from this analysis do have merit**, the authors' argument for "germline screening protocols modified based on ethnicity" is not well supported by the current study.

1. Descriptive Statistics of Clinical and Molecular Features

- a. The authors describe a bimodal age distribution for breast cancer incidence in the EUR population and unimodal distributions in the AFR and SAS populations. However, it appears that all three populations may have overlapping Gaussian distributions and may all be multimodal.
- b. Detailed attention is given to measures of deprivation. However, this measure does not feature heavily in the main analyses of the paper other than its inclusion as a covariate. It could be incorporated into the analyses for somatic mutations and somatic mutational signatures.
- c. Minor: The discussion on allostatic load and breast cancer features needs to be supported by references from the literature.

a. We thank the reviewer for this point and make the observation that the apparent bimodality of some of the algorithms may be an artefact of the smoothing algorithm used for plotting the distributions. While the AFR and SAS (Black and South Asian for BCN Biobank) age distributions are unimodal, the EUR age distribution for Genomics England could show some bimodality.

b. We have re-run the analysis for the somatic mutations, including IMD as a factor, which, although altering the p-values of the differentially-present mutations in our comparisons slightly, did not change the variants which were significant.

c. We have added a discussion on allostatic load and overall cancer mortality (Discussion p26, 574-578).

2. Somatic Variants in the Genomics England Cohort

- a. The authors identified differences in mutational frequencies for genes in the AFR and SAS Genomics England Cohorts; 6 of these genes were also differentially mutated between AFR and EUR populations in TCGA. However, it is unclear whether the comparison to the TCGA cohort is based on results from a logistic or threshold model.

Further the presentation of both approaches for the GEC cohort makes it somewhat difficult to follow the text. The authors appear to emphasize the logistic approach and thus may consider presenting these results only to improve clarity.

We have amended the text to ensure clarification on the reasons used for each model and justification of providing a union of the models.

The TCGA variant data was used as a validation rather than a discovery cohort, to check whether the genes and variants were altered in the non-EUR TCGA cohorts and did not assess any significance or differential presence. For this we used previously called simple somatic variants from TCGA that were available through the Genomic Data Commons of the NIH.

b. Differential mutation frequencies were identified among 7 variants in the AFR cohort and 4 variants in the SAS cohort from GEC. Figure 5 could benefit from labeling to indicate areas of the graph referring to genes vs. variants.

We have updated the figure (now Revised Figure 4) to emphasise the split between genes and variants.

3. Germline Variant Profiles of Cancer Susceptibility Genes + Validation

a. The authors state that 6 genes that were differentially mutated between the EUR and AFR cohorts were also differentially mutated in TCGA. The six genes mentioned are not featured within a primary table or figure for the manuscript but are included in supplementary text. This makes it difficult to follow the author's interpretation of the findings. Further, the authors toggle between discussions for somatic and germline mutations and should clearly state which they are referring to in the relevant text (see page 19, first paragraph).

We have the text and figure to ensure clarification between somatic and germline variants identified. Revised Figure 7 now shows the somatic and germline variants identified for pharmacogenomic potential in separate plots, clarifying the presentation.

Response to Reviewers' comments

Reviewer #1 (Remarks to the Author):

EDITORIAL NOTE: This Reviewer was not able to submit their report on this occasion. However, Reviewer #4 agreed to comment on their behalf.

Reviewer #2 (Remarks to the Author):

The answers provided by the authors are clear and satisfactory. In my opinion, the paper might be accepted.

Reviewer #3 (Remarks to the Author):

EDITORIAL NOTE: This Reviewer was not able to submit their report on this occasion. However, Reviewer #2 evaluated your response to Reviewer #3 and considered all concerns to be addressed.

Reviewer #4 (Remarks to the Author):

The authors have largely addressed all of my previous comments regarding this manuscript. I appreciate the detailed explanation of genetic ancestry within the cohorts, particularly as it pertains to admixture, as well as the interesting comparison of HRdetect between WGS and WXS in TCGA. I additionally think that the updated mutational burden analysis incorporating a negative binomial model and expanded gene set for the germline variant profiles are improvements.

A minor recommendation would be, pertaining to my comment #26, to add the reference justifying this cut-off value to the manuscript body itself.

A reference to the paper citing the p-value cut off has been added to the manuscript. We have also taken the opportunity to rectify a reference which did not collate correctly.